# Depletion or cleavage of cohesin during anaphase differentially affects chromatin structure and segregation

Jonay Garcia-Luis[1], Hélène Bordelet[2], Agnès Thierry[2], Romain Koszul[2]*, Luis Aragon[1]*

[1]DNA motors Group, MRC London Institute of Medical Sciences, London, United Kingdom; [2]Institut Pasteur, CNRS UMR 3525, Université Paris Cité, Unité Régulation Spatiale des Génomes, Paris, France

**Abstract** Chromosome segregation requires both the separation of sister chromatids and the sustained condensation of chromatids during anaphase. In yeast cells, cohesin is not only required for sister chromatid cohesion but also plays a major role determining the structure of individual chromatids in metaphase. Separase cleavage is thought to remove all cohesin complexes from chromosomes to initiate anaphase. It is thus not clear how the length and organisation of segregating chromatids is maintained during anaphase in the absence of cohesin. Here, we show that degradation of cohesin at the anaphase onset causes aberrant chromatid segregation. Hi-C analysis on segregating chromatids demonstrates that cohesin depletion causes loss of intrachromatid organisation. Surprisingly, tobacco etch virus (TEV)-mediated cleavage of cohesin does not dramatically disrupt chromatid organisation in anaphase, explaining why bulk segregation is achieved. In addition, we identified a small pool of cohesin complexes bound to telophase chromosomes in wild-type cells and show that they play a role in the organisation of centromeric regions. Our data demonstrates that in yeast cells cohesin function is not over in metaphase, but extends to the anaphase period when chromatids are segregating.

*For correspondence:
romain.koszul@pasteur.fr (RK);
luis.aragon@ic.ac.uk (LA)

**Competing interest:** The authors declare that no competing interests exist.

## Editor's evaluation

Previous studies suggest that multiple activities of cohesin, essential in mitotic chromosome structure and segregation, are required only prior to the onset of chromosome segregation, but this study in *Saccharomyces cerevisiae*, using different alleles of the Mcd1 subunit, shows that cohesin plays also a role in anaphase organizing the centromeric regions, providing new evidence that cohesin function is critical during and after the onset of chromosome segregation. The work is of relevance to understanding chromosome biology and cell division.

## Introduction

The establishment and maintenance of sister chromatid cohesion following DNA replication is a requirement for the equal segregation of chromosomes during cell division. Cohesin is a multisubunit protein complex that holds sister chromatids together (*Guacci et al., 1997*; *Michaelis et al., 1997*) and promotes condensation of metaphase chromosomes (*Guacci et al., 1997*). The core proteins of cohesin are a pair of structural maintenance of chromosomes (SMC) proteins, Smc1 and Smc3, and the conserved kleisin factor Mcd1/Scc1. Together they form a tripartite complex that has a rod-shaped structure (*Bürmann et al., 2019*). Cohesin complexes also contain the HEAT-repeat proteins Scc3 and Pds5 (*Hartman et al., 2000*; *Losada et al., 2000*). In addition to its role in sister chromatid

cohesion, studies have shown that cohesin also organises chromosomes through the formation of chromatin loops in interphase nuclei (*Rao et al., 2017*) and mitotic chromatids (*Lazar-Stefanita et al., 2017*; *Schalbetter et al., 2017*). The role of cohesin in nuclear organisation is thought to depend on its ability to extrude DNA loops, an activity that was recently confirmed in vitro with human cohesin (*Davidson et al., 2019*; *Kim et al., 2019*).

Separase cleaves cohesin's kleisin Mcd1/Rad21/Scc1 subunit (thereafter referred to as Mcd1) in mitosis (*Uhlmann et al., 1999*). Engineered Mcd1 cleavage in metaphase, using TEV protease expression and TEV recognition sites in Mcd1, triggers bulk separation of the genome (*Uhlmann et al., 2000*), demonstrating that one of cohesin's functions is to hold sister chromatids together prior to their separation. Recent work using chromosome capture (Hi-C) techniques has demonstrated that cohesin mediates intrachromosomal loops that are responsible for compacting yeast chromosome arms in metaphase cells, independently of cohesin's role in sister chromatid cohesion (*Lazar-Stefanita et al., 2017*; *Schalbetter et al., 2017*).

In mammalian cells cohesin is responsible for the organisation of the genome into loops during interphase (*Rao et al., 2017*). During prophase the bulk of cohesin dissociates from chromosome arms (*Sumara et al., 2000*), however since sister chromatids remain paired, it is likely that some cohesin complexes are retained on chromosomes. At metaphase, mammalian cohesin is mainly localised to centromeres where it provides cohesion (*Sumara et al., 2000*), while the related SMC complexes, condensin I and II (CI and CII), take over the role of folding mitotic chromosome arms into loops (*Gibcus et al., 2018*). CII organises large (200–400 kb) loops, while CI sub-divides these into smaller loops (80 kb) (*Gibcus et al., 2018*). During interphase CI is excluded from the nucleus and CII, although nuclear, is prevented from acting on chromosomal DNAs by a nuclear protein, Mcph1 (*Yamashita et al., 2011*). Interestingly, a rapid switch from condensin binding to cohesin binding between anaphase and cytokinesis has been reported (*Abramo et al., 2019*). However, the function of this chromosome-bound cohesin in telophase cells is currently unknown.

Yeast cells lack a CII complex, and CI role in chromosome organisation in mitosis is restricted to the ribosomal gene array (*Lazar-Stefanita et al., 2017*; *Schalbetter et al., 2017*). Instead, yeast condensin is involved in the separation of chromosome arms (*Leonard et al., 2015*) through a role that promotes decatenation of sister chromatids (*Baxter et al., 2011*; *Sen et al., 2016*).

The demonstration that yeast cohesin organises loops on the arms of metaphase chromosomes (*Lazar-Stefanita et al., 2017*; *Schalbetter et al., 2017*) together with the limited effect observed for condensin on yeast chromosome condensation (*Lazar-Stefanita et al., 2017*; *Schalbetter et al., 2017*) and the cleavage of cohesin by separase at the anaphase onset (*Uhlmann et al., 2000*) present a conundrum currently unexplained; how is the organisation/compaction of yeast chromatids maintained as they are segregating during anaphase? Here, we have investigated this question. We have studied the organisation of chromosome arms during segregation while we compromised cohesin function using two contrasted experimental approaches: auxin-mediated degradation and TEV-mediated cleavage of Mcd1. We demonstrate that degradation of Mcd1 severely disrupts chromosome structure and consequently the segregation of chromatids in anaphase. We demonstrate that similarly to mammalian cells, a small population of cohesin complexes are present on segregating chromatids during anaphase/telophase and show that they are important for centromere organisation. Together, these findings demonstrate that yeast cohesin has a role on segregating chromatids during anaphase.

## Results
### Degradation of cohesin's kleisin at the anaphase onset causes mitotic catastrophe

To study the role of cohesin on chromosome organisation during anaphase, we searched for an approach to induce the rapid removal of cohesin's kleisin subunit, Mcd1. To this aim, we used an auxin-inducible degron allele of Mcd1 (*MCD1-AID*) (*Nishimura et al., 2009*) which allows rapid degradation by poly-ubiquitylation upon exposure to auxin.

To evaluate the effect of Mcd1 degradation during chromatid segregation, we blocked cells using transcriptional depletion of Cdc20 (an activator of the anaphase-promoting complex). Under this experimental condition, cells are arrested with chromosomes bipolarly attached to mitotic spindles.

Removal of cohesin by artificial cleavage of Mcd1 in Cdc20-depleted cells triggers an anaphase-like stage where chromatids segregate to opposite cell poles, despite the fact that cells are biochemically in metaphase (*Uhlmann et al., 2000*).

In cdc20-arrested cells, we observed full degradation of Mcd1-aid 30–60 min after addition of auxin (*Figure 1a*), demonstrating the rapid and efficient removal of Mcd1 in this experimental setup. Like TEV-induced anaphases, auxin-mediated degradation of Mcd1 in cdc20-arrested cells precipitates an anaphase-like stage (*Figure 1b*), however we observed severe disruption of nuclear segregation (*Figure 1b*), with many nuclei appearing to be stuck in anaphase with elongated nuclear masses for extended periods (*Figure 1b–c*). To quantify the segregation defects in these cells, we introduced chromosome tags at different genomic locations, and scored the timing and efficiency of their separation. First, we used tags on the arm (*tetO::469*) and telomere (*tetO::558*) regions of chromosome 5, a small chromosome in the yeast genome (*Figure 1c–d*). After 180 min of auxin addition to cdc20-blocked cultures, we observed that 48% of the cells were still stuck in anaphase (*Figure 1c*), 35% of cells showed correct segregation of arm tags, and 20% showed missegregation (*Figure 1c*). Segregation errors were even higher for telomeric regions (*Figure 1d*) and larger chromosomes (*Figure 1—figure supplement 1*). These results demonstrate that rapid removal of Mcd1 by degradation causes an anaphase-like state with impaired separation of chromatids.

## TEV-induced cleavage allows bulk genome separation with minor segregation errors

Previous studies have shown that engineered cleavage of Mcd1 in cdc20 arrests, using TEV protease expression and TEV recognition sites on Mcd1, triggers an anaphase-like division where nuclear masses separate (*Uhlmann et al., 2000*). We re-evaluated segregation in TEV-induced anaphases using the previously published strain and protocol (*Uhlmann et al., 2000*). Induction of TEV expression led to cleavage of Mcd1 after 60 min (*Figure 1—figure supplement 2*) as expected. Bulk nuclear segregation occurred 90–120 min following the induction as it had been previously reported (*Uhlmann et al., 2000*). Importantly, chromosome tags located on the centromere region of chromosome 5 segregated efficiently (*Figure 1—figure supplement 2*). Therefore, unlike auxin-mediated degradation of Mcd1, and consistent with previous reports (*Uhlmann et al., 2000*), TEV cleavage allows nuclear segregation.

We noticed that in TEV-induced anaphases, the cleaved C-terminal fragment of Mcd1 was fully stable during the entire timecourse (*Figure 1—figure supplement 2*). The lack of Mcd1 fragment degradation after TEV cleavage stems from the fact that TEV protease cleavage occurs following the glutamine (Q) residue of the TEV recognition site ('ENLYFQ*G') leaving a glycine (G) amino acid residue at the N-termini (referred to as TEVG), which is not well recognised by the N-rule pathway (*Varshavsky, 2011*). In contrast, separase cleavage leaves an arginine (R) residue at the N-termini of the cleaved product ('SVEQGR*R'), which is a good substrate for N-end rule degradation (*Varshavsky, 2011*). Interestingly, Beckouet et al. have shown that following TEV-induced cleavage not only the C-terminal fragment of Mcd1 is stabilised but also the N-terminal (*Beckouët et al., 2016*). This raises the possibility that Mcd1 fragments could remain associated to the Smc core subunits following TEV cleavage. To test whether this is the case, we tagged Smc3 with the V5 epitope and Mcd1 with FLAG and HA tags at the N- and C-terminus, respectively (*Figure 2*). We performed IPs on synchronised TEV anaphases (*Figure 2*) to follow whether the cleaved fragments stay associated to the cohesin Smc core. Both N- and C-terminal Mcd1 fragments were immunoprecipitated by Smc3 after TEV cleavage (*Figure 2—figure supplement 1*). Therefore, the structural integrity of cohesin tripartite complex remains intact after TEV cleavage of its kleisin subunit. Next, we sought to test whether inducing full degradation of Mcd1 fragments after TEV cleavage generates a similar phenotype to that observed in anaphases induced by degradation of Mcd1 (*Figure 1c*).

To this aim, we used a TEV recognition site on Mcd1 that was able to yield a C-terminal fragment with a terminal amino acid recognised by the N-end rule pathway. We found this to be the case when we used the TEV recognition site 'ENLYFQF' (referred to as TEVF) (*Figure 3a*). This site leaves a phenylalanine, rather than a glycine, as the N-terminal amino acid after TEV cleavage. Importantly, N-terminal phenylalanine is a good substrate for N-end rule degradation (*Varshavsky, 2011*). We used FLAG and HA tags at the N- and C-terminus of Mcd1, respectively, to detect the two products generated by TEV cleavage and compared their stability in Mcd1 proteins containing the classical

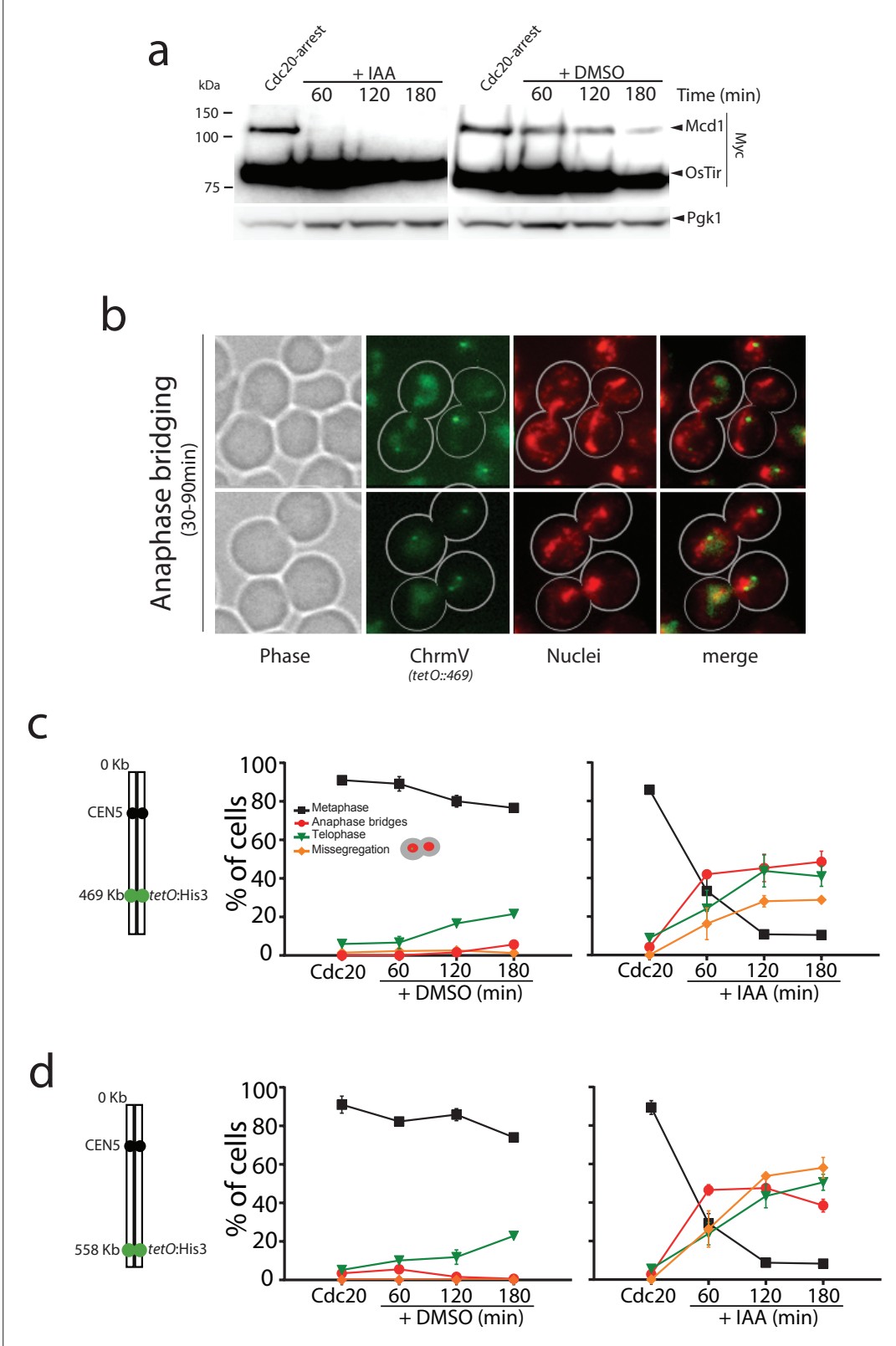

**Figure 1.** Mcd1 degradation causes catastrophic chromosome segregation. (**a**) Cells containing *MCD1* tagged with the auxin degron (*MCD1-AID*) were arrested in metaphase (Cdc20 arrest). The culture was split in two, one half was treated with DMSO and the other with 6 mM auxin (IAA) to degrade Mcd1. Samples were taken for an anti-Myc immunoblotting to detect Mcd1. (**b**) Representative images of cells 30–90 min after degradation of Mcd1.

*Figure 1 continued on next page*

*Figure 1 continued*

Cells were analysed for nuclear separation (DAPI stain, red) and chromosome segregation (green fluorescent protein [GFP] dots marking the middle of chromosome V; *tet:469*, green). (**c**) Analysis of nuclear and chromosome segregation using DAPI and GFP dots marking the middle of chromosome V (*tet:469*). Experimental conditions for the timecourse are as described in (a). Each timepoint represents the average of the percentage of three biological replicas of at least 100 cells per timepoint. Error bars show SEM. (**d**) Analysis of nuclear and chromosome segregation using DAPI and GFP dots marking the telomeric region of chromosome V (*tet:558*). Experimental conditions for the timecourse are as described in (a). Each timepoint represents the average of the percentage of three biological replicas of at least 100 cells per timepoint. Error bars show SEM.

The online version of this article includes the following source data and figure supplement(s) for figure 1:

**Source data 1.** Source data 1 contains two blots shown in main *Figure 1a*.

**Figure supplement 1.** Mcd1 degradation causes catastrophic segregation of large chromosomes.

**Figure supplement 1—source data 1.** One blot shown.

**Figure supplement 1—source data 2.** One blot shown.

**Figure supplement 2.** TEV-mediated cleavage of Mcd1 allows nuclear segreagation.

**Figure supplement 2—source data 1.** Source data 2 contains one blot shown in *Figure 1—figure supplement 2*.

---

'ENLYFQG' (TEVG) or 'ENLYFQF' (TEVF) sites (*Figure 3a–b*). Cleavage of TEVG generated stable N- and C-terminal fragments as expected (*Figure 3b*). In contrast, both N- and C-terminal fragments were rapidly degraded after cleavage on TEVF sites (*Figure 3a*).

Next, we compared segregation kinetics in TEV-induced anaphases with TEVG and TEVF cleavage sites. Bulk nuclear separation was observed in both conditions (*Figure 3a–b*), with most cells showing full segregation after 120 min of TEV induction (>85% in TEVG and >75% in TEVF). However, we noticed minor delays during anaphase progression in cells with TEVF recognition sites (*Figure 3a*). Next, we compared the fidelity of segregation in TEVG and TEVF anaphases using tetO:469 kb tags inserted in the middle of chromosome 5. As observed previously, cells carrying TEVG recognition sites segregated tags correctly (with <5% missegregation observed) (*Figure 3c*). In contrast, tag missegregation was observed in 16% of telophases when TEVF recognition sites were present on Mcd1 (*Figure 3c*). These results demonstrate that degradation of Mcd1 fragments after TEV cleavage affects the fidelity of chromosome segregation but, unlike Mcd1 degradation, does not severely prevent bulk nuclear separation.

## Chromosome looping in segregating chromosomes is disrupted in the absence of cohesin

Cohesin mediates intrachromosomal loops in metaphase-arrested chromosomes, providing a structural framework for the compaction of chromosome arms (*Lazar-Stefanita et al., 2017*; *Schalbetter et al., 2017*). How chromosomes are organised during yeast anaphase, when individual chromatids are being pulled to the poles, is not well understood. Since the cohesin tripartite complex remains associated after TEV cleavage (*Figure 2*), we considered the possibility that TEV-cleaved cohesin still plays a role in maintaining the structure of segregating chromatids. To further investigate this, we first performed ChIP analysis of Smc3 during TEV-induced anaphases along chromosome 5 (*Figure 4a*). We observed that significant levels of chromatin-bound Smc3 remained during initial timepoints (*Figure 4a*; TEV-induced). In contrast, Smc3 was rapidly lost from chromatin in anaphases induced by auxin-mediated degradation of Mcd1 (*Figure 4a*; auxin-induced). Having observed that TEV-cleaved cohesin remains chromatin-bound for a significantly longer period than auxin-degraded cohesin (*Figure 4a*), we decided to use these two conditions to investigate whether the presence of cohesin, albeit cleaved, impacts on the looped organisation of chromosomes as they are pulled to the cell poles. We built Hi-C libraries from timepoints when the bulk of Mcd1 had been either TEV-cleaved or auxin-degraded (*Figure 4—figure supplement 1*). Importantly, we did not detect full-length Mcd1 in the TEV-cleaved samples collected (*Figure 4—figure supplement 1*), suggesting that TEV cleavage was complete. Following sequencing, we computed the corresponding normalised genome-wide contact

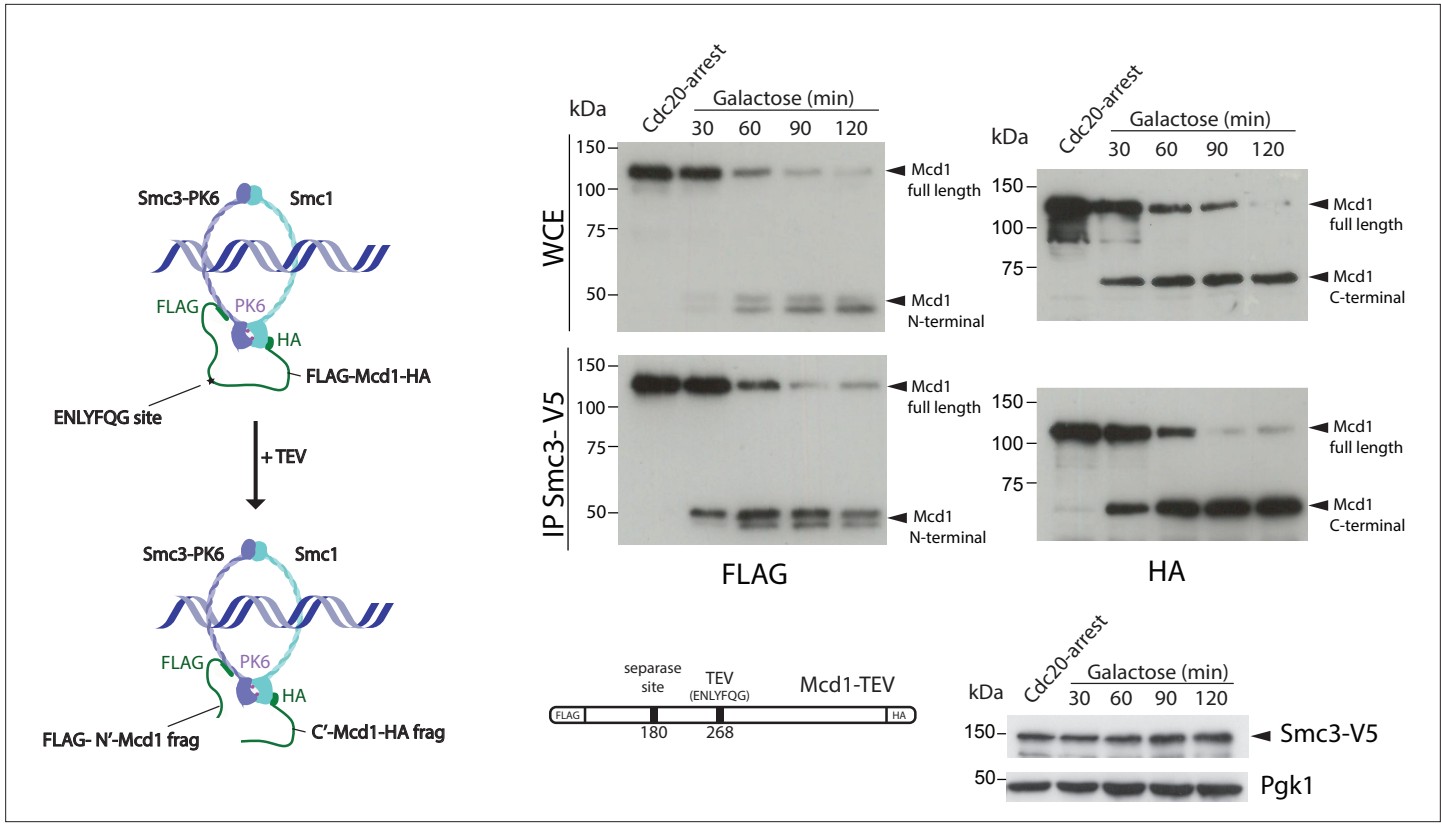

**Figure 2.** Cohesin ring structure remains after tobacco etch virus (TEV) cleavage of Mcd1. Schematic of engineered cohesin with *MCD1* tagged in N-terminus with FLAG, the C-terminus with HA, and with a substitution of the 268 separase cleavage site with the TEV cleavage sequence ENLYFQG. Cells also harboured a copy of Smc3 tagged in C-terminus with V5 (left). Cells were synchronised using a Cdc20 arrest and TEV was induced to cleave Mcd1. Samples were taken every 30 min for 2 hr and Smc3 immunoprecipitated using anti-V5 antibody. We used immunoblotting with anti-FLAG and HA antibodies to detect the cleaved fragments of Mcd1 (right).

The online version of this article includes the following source data and figure supplement(s) for figure 2:

**Source data 1.** Six blots shown in main *Figure 2*.

**Figure supplement 1.** Cells with *MCD1* tagged in N-terminus with FLAG, C-terminus with HA, with the 268 separase cleavage position substituted by the tobacco etch virus (TEV) cleavage sequence ENLYFQG (TEV-G) and with *SMC3* tagged in C-terminus with V5 were synchronised in metaphase (Cdc20) and Mcd1 was cleaved in the TEV cleaving site.

**Figure supplement 1—source data 1.** Three blots shown.

**Figure supplement 1—source data 2.** Two blots shown.

maps for Mcd1-TEV and Mcd1-aid (*Figure 4b–c*). When we compared the contact maps of individual chromosomes obtained from cells arrested in metaphase using Cdc20 depletion (*Figure 4b*), to those obtained for Mcd1-TEVG and Mcd1-AID during the induced anaphases, we observed a decrease in intrachromosomal contacts structuration, with a loss of loops (*Figure 4b*). The contact probability (p) as a function of genomic distances of all chromosome arms showed a reduction of contacts in the 10–50 kb range for Mcd1-TEVG and Mcd1-AID samples compared to Cdc20 arrests (*Figure 4c*, *Figure 4—figure supplement 1*). Notably, the reduction in Mcd1-AID was significantly more pronounced than in Mcd1-TEVG (*Figure 4c*, *Figure 4—figure supplement 2*). These results demonstrate that removal of cohesin from chromatin in Mcd1-AID causes a very pronounced disruption of cohesin-dependent structure (*Figure 4b–c*) that prevent correct chromosome organisation, and thus explain the catastrophic missegregation observed (*Figure 1c–d*). In contrast, chromosome organisation is not fully disrupted when TEV-cleaved, but chromatin-bound, Mcd1 (and cohesin) is present (*Figure 4c*, *Figure 4—figure supplement 1*), which is likely to maintain a level of structure (albeit reduced) that allows bulk nuclear segregation (*Figure 3a–c*).

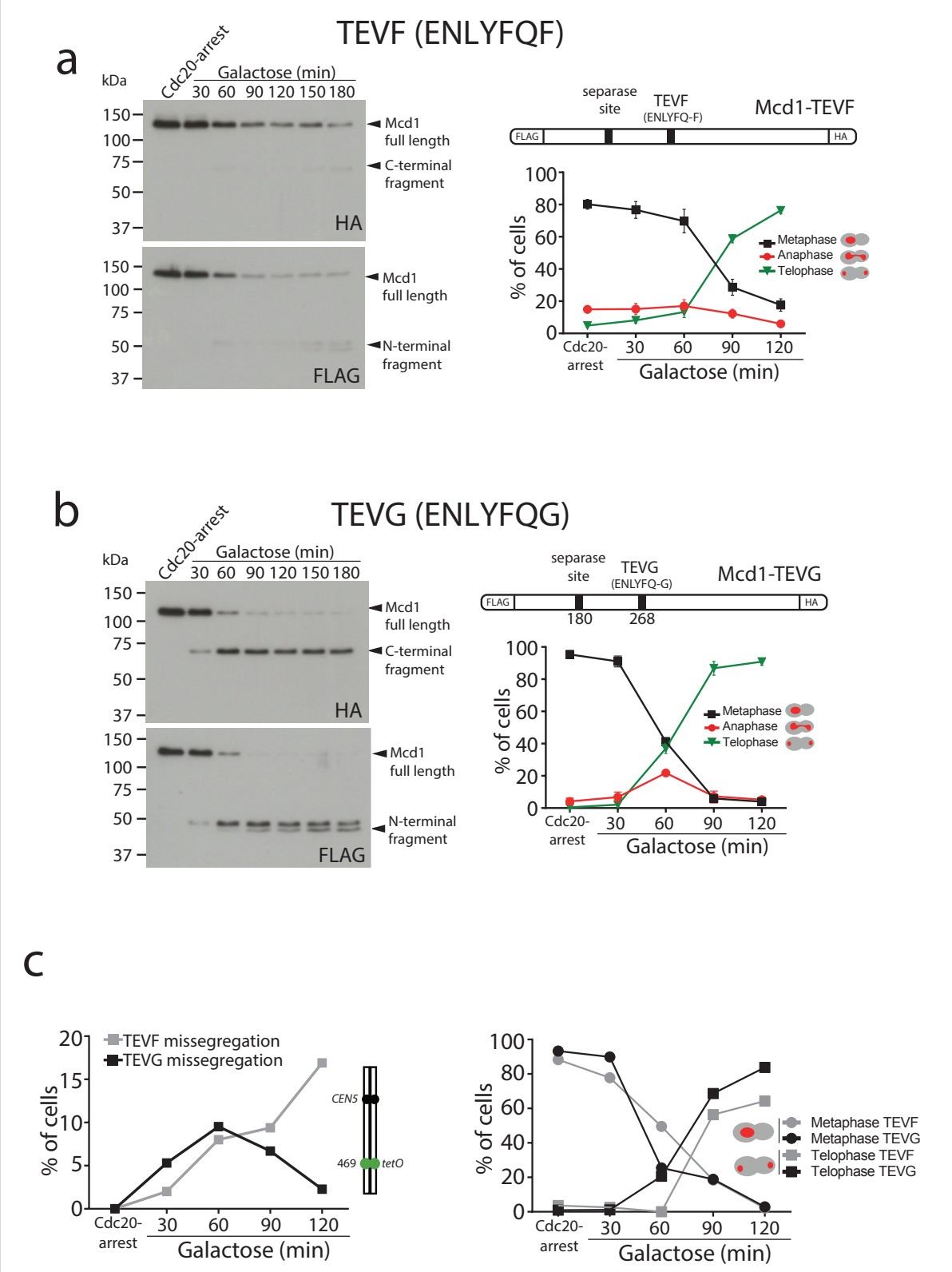

**Figure 3.** Degradation of Mcd1 fragments after tobacco etch virus (TEV) cleavage affects segregation efficiency. (a) Cells with *MCD1* tagged at its N-terminus with FLAG and its C-terminus with HA, and with the 268 separase cleavage site replaced by the TEV recognition site ENLYFQF (TEVF) (top right diagram) were arrested in Cdc20 before TEV induction. Samples were taken every 30 min for 2 hr and immunoblotted against FLAG or HA to detect Mcd1 N-terminus and C-terminus cleaved fragments, respectively (left). Nuclear segregation was monitored during the TEV-induced anaphase with

*Figure 3 continued on next page*

*Figure 3 continued*

DAPI staining (bottom right graph). Error bars are standard deviations. (**b**) Cells with *MCD1* tagged at its N-terminus with FLAG and its C-terminus with HA, and with the 268 separase cleavage site replaced by the TEV recognition site ENLYFQG (TEVG) (top right diagram) were arrested in Cdc20 before TEV induction. Samples were taken every 30 min for 2 hr and immunoblotted against FLAG or HA to detect Mcd1 N-terminus and C-terminus cleaved fragments, respectively (left). Nuclear segregation was monitored during the TEV-induced anaphase with DAPI staining (bottom right graph). Error bars are standard deviations. (**c**) Cells carrying *MCD1* with the 268 separase cleavage site substituted for either TEV cleavage ENLYFQF (TEVF) or ENLYFQG (TEVG) were treated as in (a and b) and monitored for nuclear and chromosome segregation using DAPI and green fluorescent protein (GFP) dots marking the middle of chromosome V (*tet:469*). Each timepoint represents the average of the percentage of three biological replicas of at least 100 cells per timepoint. Error bars show SEM.

The online version of this article includes the following source data for figure 3:

**Source data 1.** Two blots shown in main *Figure 3a*.

**Source data 2.** Two blots shown in main *Figure 3b*.

## Cohesin organises centromeres of telophase-arrested chromosomes

Our results demonstrate that the fidelity of chromosome segregation requires the maintenance of cohesin-dependent looping during anaphase. Next, we investigated whether cohesin complexes are removed in fully segregated chromosomes when they reached the cell poles in telophase. To this aim, we first fused the green fluorescent protein (GFP) to the cohesin subunit Smc3 and imaged its localisation on telophase-arrested cells using the *cdc15-2* conditional mutant (*Jaspersen et al., 1998*; *Figure 5a*). Smc3-GFP signal was observed on segregated nuclei, with a clear discrete dot present at the cell poles that colocalised with the spindle pole body protein Spc29 (Spc29-RedStar2) (*Figure 5a*). This indicates that Smc3-GFP might be enriched on centromeric regions of telophase chromosomes. To confirm this possibility, we used calibrated ChIP-seq in *cdc15-2*-arrested cells (*Figure 5b*, *Figure 5—figure supplement 1*) to identify whether centromeric regions and any other potential genomic sites are bound by cohesin's subunit Mcd1. We performed CHIP analysis using *MCD1-6HA* in *cdc15-2*-arrested cells (*Figure 5b*). To have a visual reference for the normal position of cohesin sites on chromosomes, we used a previously published dataset (*Garcia-Luis et al., 2019*) for Smc1 localisation on cells arrested in metaphase where cohesin binding is maximal (*Figure 5b*; *SMC1* reference). To ensure that signals detected in telophase arrests were not due to background noise, we subtracted the signal obtained using untagged cells in our analysis (*Figure 5b*; untagged substracted). The number of Mcd1 binding sites along chromosome arms was very low, with only a few sites exhibiting levels above background (*Figure 5b*). However, comparison of average MCD1-6HA profiles across *CEN* sites confirmed that this cohesin subunit is enriched around centromere regions in *cdc15-2*-arrested chromosomes (*Figure 5c*, *Figure 5—figure supplement 2*), thus validating our previous cytological observations (*Figure 5a*).

Next, we sought to test whether centromere-bound cohesin contributes to the organisation of these regions in telophase. We arrested cells using an analogue-sensitive (AS) allele of Cdc15, and inactivated cohesin using the Mcd1-AID and Smc3-AID alleles after telophase arrest had been achieved (*Figure 6a*). We then built Hi-C libraries from cells arrested in telophase with and without degrading cohesin after the arrest (*Figure 6b*, *Figure 6—figure supplement 1*). Comparison of the contact maps revealed changes at centromeric regions in telophase cells depleted for cohesin (*Figure 6b*). On a large scale (above 200 kb), cohesins favour *cis* contacts between centromeres and their chromosome arms (*Figure 6b and c* top). However, this effect is reversed at short distances (<100 kb). Indeed, panels in *Figure 6c* show that cohesin impedes contacts between centromeres and their ~100 kb flanking regions. These results demonstrate that cohesin complexes influence intrachromatid contacts at centromeres in telophase chromosomes. Moreover, the in trans interaction of *CEN* sequences was also reduced in *cdc15-as* cells with depleted cohesin (*Figure 6d*). Therefore, inactivating cohesin in telophase also reduces centromere clustering of the yeast genome.

Cohesin also contributes to the organisation of the ribosomal gene array (rDNA) on chromosome XII during metaphase (*Lavoie et al., 2002*). We therefore tested whether inactivation of cohesin in telophase-arrested cells had any effects on rDNA structure. To this aim we used an AS allele of *cdc15-as* and the temperature-sensitive allele of cohesin's kleisin *mcd1-73*. We expressed the nucleolar marker Net1 fused to GFP (*NET1*-yeGFP) in these cells to evaluate rDNA structure. Inactivation of cohesin in telophase arrests caused decondensation of rDNA signals (*Figure 7a*).

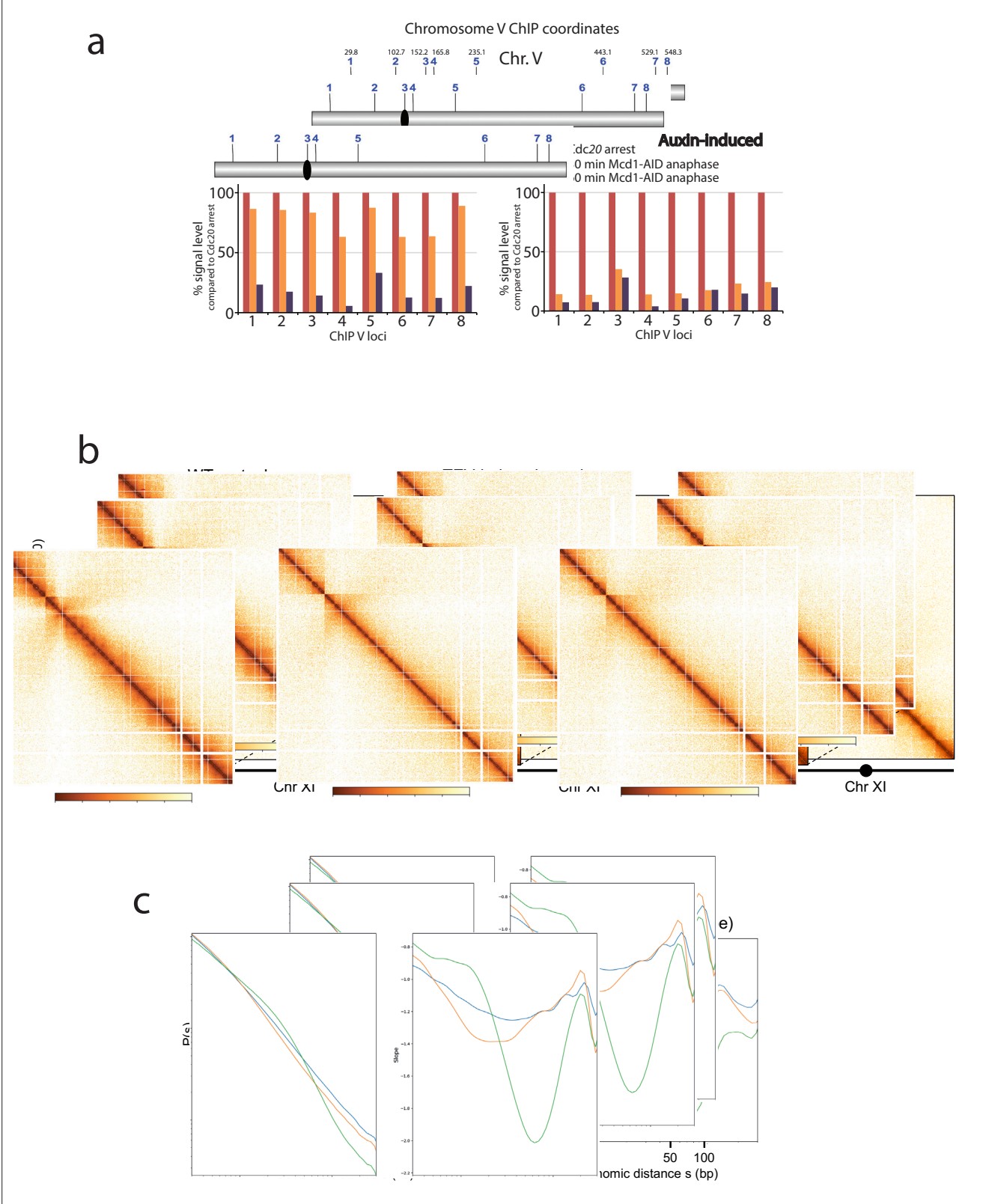

**Figure 4.** Depletion or cleavage of cohesin differentially affects chromatin structure during anaphase. (**a**) Chromatin immunoprecipitation analysis (ChIP) of Smc3-V5 binding along chromosome V of cells arrested in metaphase (Cdc20 arrest) containing either *MCD1* with the 268 separase cleavage site substituted by the tobacco etch virus (TEV) recognition sequence ENLYFQF (TEVF) or with *MCD1-AID*. Samples were taken every 30 min for 1 hr after induction of the TEV protease or addition of the auxin IAA respectively and analysed. (**b**) Cells containing either *MCD1-TEVG or MCD1-AID*

*Figure 4 continued on next page*

*Figure 4 continued*

were arrested in Cdc20 metaphase arrest and MCD1 was cleaved or degraded respectively. Samples for HiC analysis were taken (*MCD1 TEVG* 90 min; *MCD1-AID* 60 min). Contact maps (bin = 1 kb) of chromosome V from cell populations are shown. Brown to yellow colour scales represent high to low contact frequencies, respectively (log10). A Cdc20 metaphase arrest was also processed as a reference. n=1 (left) and n=2 (middle and right) biological replicates for each condition. (**c**) Average intrachromosomal arm contact frequency (p) between two loci with respect to their genomic distance (**s**) along the chromosome, of cell populations treated as in 'a' (left). Derivative of p(s) curve (right).

The online version of this article includes the following source data and figure supplement(s) for figure 4:

**Figure supplement 1.** Cells containing either *MCD1 TEVG or MCD1-AID* were arrested in Cdc20 metaphase arrest and MCD1 was cleaved or degraded, respectively.

**Figure supplement 1—source data 1.** Two blots shown.

**Figure supplement 2.** Contact probability as a function of genomic distance P(s), and its derivative, determined from replicates of samples in *Figure 4c*.

Next, we looked at chromosome XII HiC contact maps of cells arrested in cdc15 and compared it to cells arrested in cdc15 with either Mcd1 or Mcd1 and Smc3 degraded (*Figure 7b*). In cdc15-arrested cells, condensin-dependent loop bridges the CEN12 and rDNA (*Lazar-Stefanita et al., 2017*). In cohesin-depleted cdc15-arrested cells, these contacts are also strongly decreased (*Figure 7b*, yellow arrow), suggesting that cohesin plays a role in the formation of this structure. Furthermore, contacts between pre- and post-rDNA regions are increased when the cohesin complex is disrupted, suggesting that it promotes the isolation of these regions in cdc15-arrested cells (*Figure 7b*). In conclusion, these results further support an active role for cohesin in the organisation of specific regions of the genome in telophase-arrested cells.

## Discussion

The role of cohesin complexes in supporting mitotic chromosome architecture in yeast has long been established (*Guacci et al., 1997*; *Michaelis et al., 1997*), however, while their prominent role in mediating sister chromatid cohesion has been intensively studied, their contribution to chromosome folding has only recently become topical with the development of Hi-C techniques (*Dekker et al., 2013*; *Schmitt et al., 2016*). Here, we set out to explore whether cohesins influence or play a role in chromosome structuration after metaphase.

Cohesin is loaded onto chromosomes by the loader complex Scc2/4 (*Ciosk et al., 2000*) during $G_1$ and becomes cohesive during DNA replication (*Uhlmann and Nasmyth, 1998*). The cohesive state requires the acetylation of cohesin Smc3 by the Eco1 acetyl transferase to make cohesin refractory to an inhibitory 'anti-establishment' activity dependent on Wapl (*Rolef Ben-Shahar et al., 2008*; *Unal et al., 2008*). In yeast, cleavage of one of cohesin subunits, Mcd1/Scc1, by the protease separase (Esp1) prompts anaphase segregation (*Uhlmann et al., 1999*; *Uhlmann et al., 2000*). Separase function is highly regulated to prevent its premature activation before every single chromosome has been aligned on the mitotic spindle. Firstly, separase activity is blocked by a bound inhibitor named securin (Pds1) (*Ciosk et al., 2000*), which is destroyed by ubiquitin-mediated proteolysis once cells are ready to enter anaphase (*Cohen-Fix et al., 1996*). Secondly, separase-mediated cleavage of Mcd1 is primed by Polo-kinase (Cdc5)-dependent phosphorylation of Mcd1 at serines 175 (S175) and 268 (S268), on the cleavage sites (*Alexandru et al., 2001*). This double regulation ensures that cohesion is not destroyed before it should be. At the anaphase onset, Mcd1 cleavage by separase is thought to terminate the function of cohesin on yeast chromosomes (*Nasmyth, 2001*). Interestingly, cohesin's subunits have been shown to be chromatin-bound during anaphase/telophase (*Renshaw et al., 2010*; *Tanaka et al., 1999*), however the functional contribution of this cohesin population has not been studied.

. Mammalian cohesin organises the genome into loops during interphase (*Rao et al., 2017*) but dissociates from chromosomes in prophase (*Sumara et al., 2000*), and it is only found at centromeres by the time cells reach metaphase (*Sumara et al., 2000*). The role of organising loops on mammalian metaphase chromosomes is taken over by CI and CII complexes (*Gibcus et al., 2018*). During chromosome segregation, CI and CII are believed to maintain the structure of separating chromatids to prevent a 'cut' phenotype, which occurs when insufficiently condensed chromatids are trapped by the cytokinetic furrow. Recent studies demonstrate that condensin and cohesin are mutually exclusive

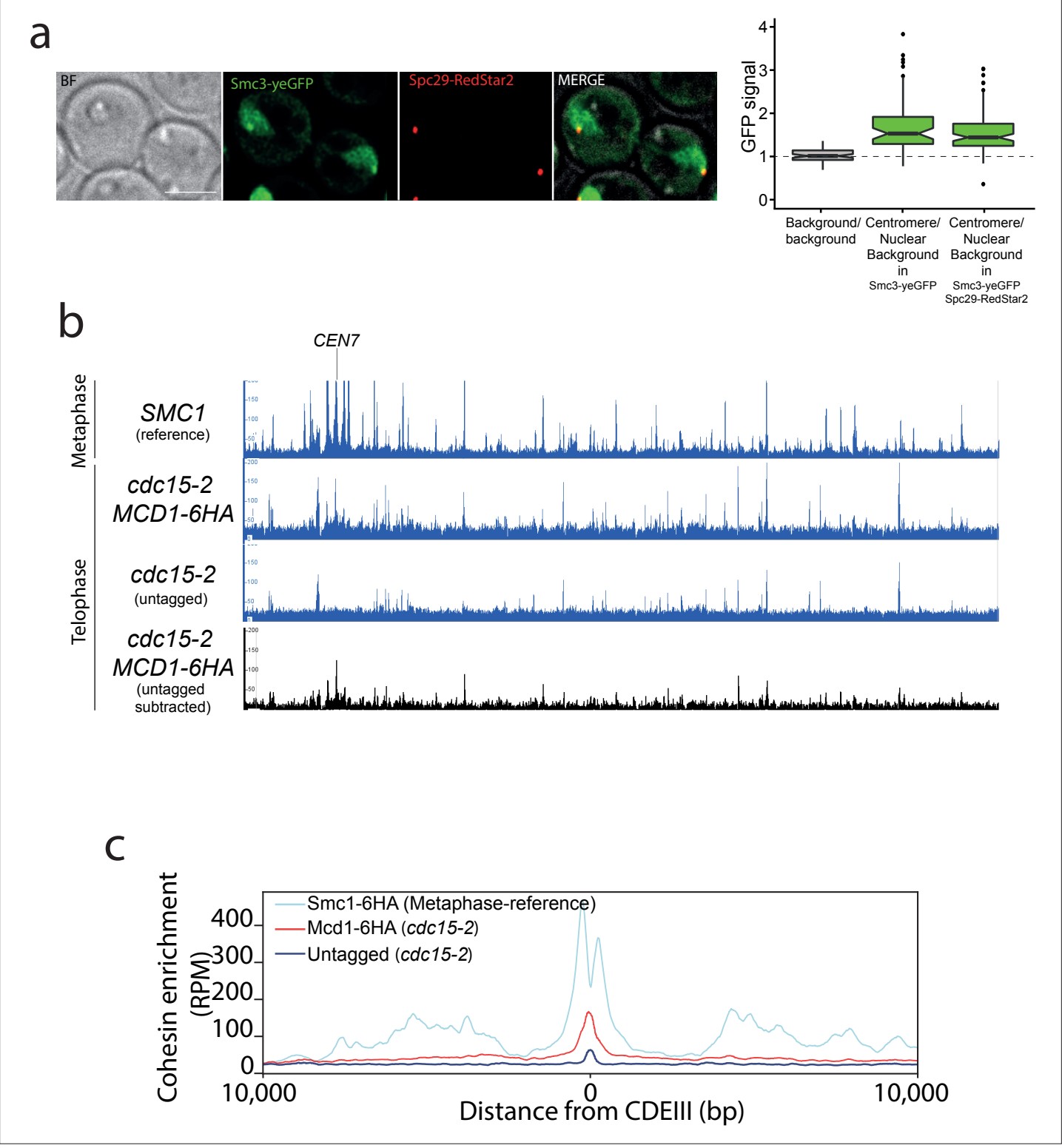

**Figure 5.** Cohesin is present around centromere regions in telophase-arrested cells. (**a**) Cells containing *CDC15-AID* and the tagged cohesin subunit *SMC3-yeGFP* were arrested in late anaphase and the green fluorescent protein (GFP) signal at the centromere was calculated as a ratio comparing it to the nuclear background signal. Cells also carried Spc29-RedStar2, a spindle pole body component that was used as spatial reference to determine colocalisation with centromeres. For each timepoint at least 20 cells of two biological replicas were quantified. (**b**) Enrichment of cohesin along *Saccharomyces cerevisiae* chromosome 7 measured by calibrated ChIP-seq in cells containing *MCD1-6HA* arrested in late anaphase (*cdc15-2*) and

*Figure 5 continued on next page*

*Figure 5 continued*

untagged cells arrested in telophase (*cdc15-2*). We used a previously published dataset (GSE118534) for Smc1 binding to metaphase-arrested cells as a reference to illustrate the position of cohesin sites on metaphase chromosomes (top; *SMC1* reference). The bottom black lane shows the enrichment of cohesin subunit Mcd1-6HA after subtraction of ChIP-seq signal of the untagged cells arrested in telophase (*cdc15-2*). *CEN7* marks the location of the centromere. (**c**) Average calibrated ChIP-seq profiles of Mcd1-6HA (telophase arrest, *cdc15-2*) from the centromere *CDEIII* region of the 16 yeast chromosomes is shown. Smc1 profile from cells arrested in metaphase is shown as a reference (GSE118534).

The online version of this article includes the following figure supplement(s) for figure 5:

**Figure supplement 1.** Microscope quantification of the cell cycle stage of cells used for ChIP-seq experiments, containing SMC1-6HA arrested in metaphase (nocodazole), untagged cells arrested in metaphase (nocodazole), cells containing MCD1-6HA arrested in *cdc15-2* late anaphase, and untagged cells arrested in *cdc15-2*.

**Figure supplement 2.** Enrichment of cohesin around CENs measured by calibrated ChIP-seq.

on mammalian chromosomes (*Abramo et al., 2019*). Interestingly, from anaphase to cytokinesis, chromosome-bound cohesin appears as condensin comes off chromosomes (*Abramo et al., 2019*). However, chromosome folding in telophase was found to be independent of both condensin and cohesin (*Abramo et al., 2019*). In yeast, there is a single condensin complex but it does not play a role in the overall organisation of chromosome loops during mitosis (*Lazar-Stefanita et al., 2017*; *Schalbetter et al., 2017*), instead it has a very defined function organising the ribosomal gene array on chromosome XII (*Lavoie et al., 2002*). Since yeast condensin does not contribute to the structural organisation of chromosome arms and cohesin's function is thought to stop at the anaphase onset, it was unclear how separating chromatids maintain the looped organisation necessary for chromosome compaction and faithful segregation.

Here, we tested the belief that cohesin has no roles after metaphase. Our results demonstrate that, in contrast to the current view, degrading Mcd1 during anaphase leads to catastrophic segregation where many cells arrest with unseparated nuclear masses (*Figure 1a–d*). This result prompted us to further investigate (i) whether the structure of segregating chromatids is disrupted when Mcd1 is degraded and (ii) whether cleavage of cohesin subunit Mcd1 causes a similar phenotype to its degradation. Hi-C analysis during segregation demonstrated that Mcd1 degradation causes dramatic defects in the 10–50 kb range of *cis* interactions on chromatids, which correspond to cohesin-dependent loop range (*Figure 4c*). Importantly, despite the exclusion of the pericentromeric regions, contacts between 20 and 200 kb remain more enriched in the Mcd1-TEV condition than in the Mcd1-AID condition suggesting that the cleaved cohesin complex still contributes, directly or indirectly, to the structuration of chromosome arms after metaphase. This data confirms that loops are affected when cohesin is not present on segregating chromatids and offers an explanation for the defective segregation observed under these conditions (*Figure 1c–d*).

To study the effect of cohesin cleavage on the efficiency and structure of segregating chromatids, we used the TEV protease system previously established (*Uhlmann et al., 2000*). Our analysis revealed that unlike Mcd1 degradation, TEV-induced cleavage does not prevent bulk nuclear separation (*Figure 3*), although it causes some segregation errors (*Figure 3c*). Importantly, using HiC we could detect the maintenance of *cis* contacts in the 10–50 kb range on chromosomes with TEV-cleaved Mcd1 (*Figure 4c*), demonstrating that chromatid loops are not fully disrupted and explaining why bulk segregation is still achieved. We ensured that TEV cleavage of Mcd1 proceeded to completion in our samples (*Figure 4—figure supplement 1*) to prevent that residual Mcd1 (that might escape cleavage) significantly affected the experimental outcome. An important difference between cohesive cohesin and loop extruding complexes lies in the way that these complexes interact with the DNA substrate. While cohesive cohesin topologically embraces DNA (*Srinivasan et al., 2018*), loop extruding complexes rely on non-topological interactions (*Davidson et al., 2019*). Therefore, it is likely that cohesin complexes cleaved by TEV are still able to interact non-topologically with the DNA substrate (*Figure 7b*) and/or can maintain extruded loops. This would explain why TEV cleavage has only a modest effect on the loop extrusion (structural) role of cohesin, while disrupting the cohesive role (that depends on topological association) (*Figure 7b*). In contrast, auxin-mediated degradation would abolish both topological and non-topological interactions (*Figure 7c*), which would result in the loss of cohesion but also structure (and compaction) thus generating a situation where unorganised chromatids would fail to efficiently segregate (*Figure 7c*).

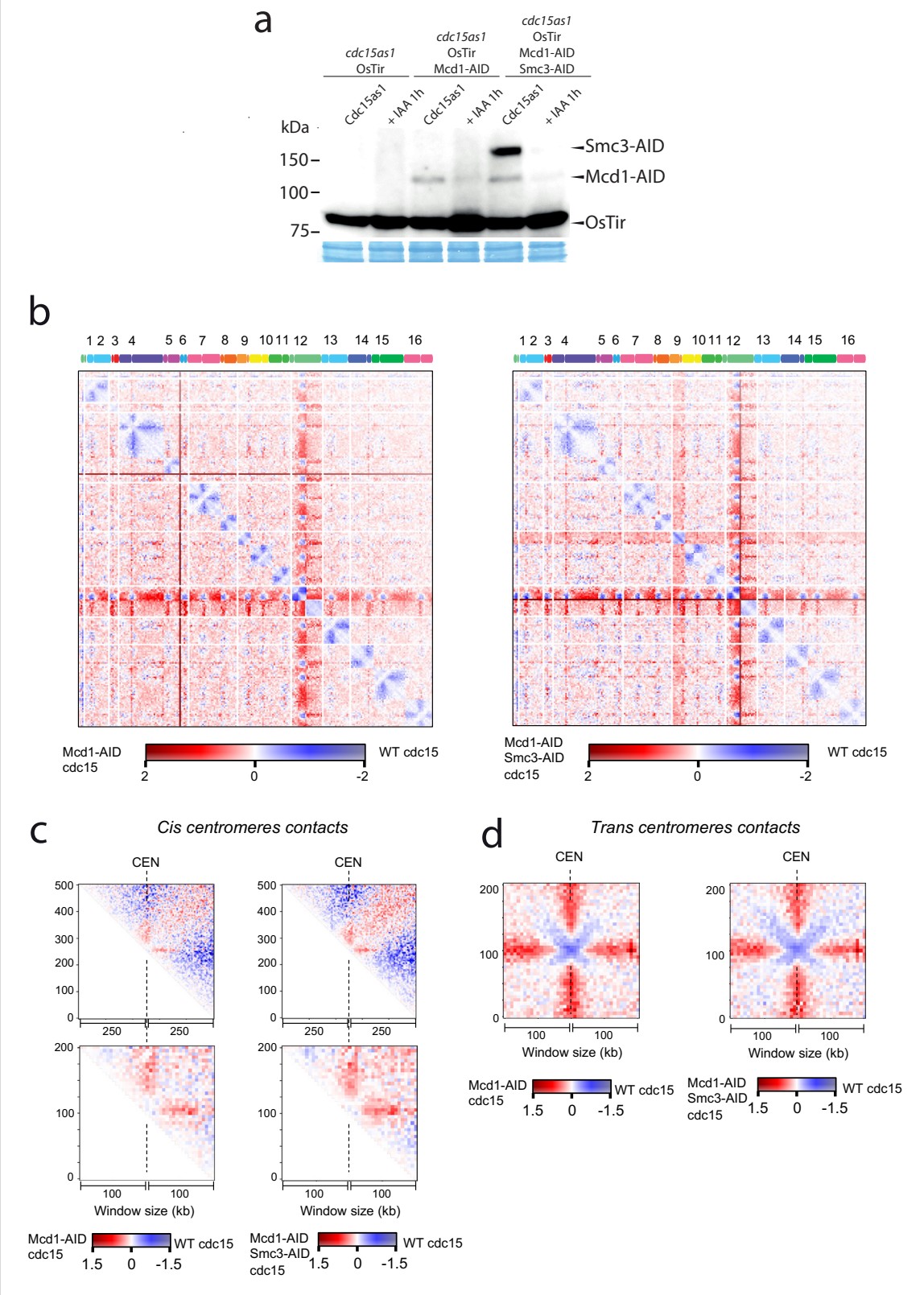

**Figure 6.** Cohesin organises centromere regions in telophase-arrested cells. (**a**) Degradation of cohesin subunits in telophase-arrested cells using *cdc15-as* allele. Cdc15-as cells and cdc15-as cells carrying MCD1-AID or MCD1-AID and SMC3-AID were synchronised in late anaphase (cdc15-as) and treated with IAA for 1 hr to deplete Mcd1-AID and Smc3-AID. Samples were taken for HiC and for immunoblotting to follow the degradation of Mcd1 and Smc3. (**b**) Log2 ratio of contact maps between *cdc15-as* arrested cells depleted in Mcd1 and *cdc15-as* arrested wild-type (WT) cells (left). Log2

*Figure 6 continued on next page*

*Figure 6 continued*

ratio of contact maps between *cdc15-as* arrested cells depleted in Mcd1 and Smc3 and *cdc15-as* arrested WT cells (right). x-axis represents the 16 chromosomes of the yeast genome depicted on top of the matrix. Blue to red colour scales represent the enrichments in contacts in one sample respect to the other (bin = 50 kb). (n=1 biological replicate for each condition). (**c**) Log2 ratio between *cdc15as* Mcd1 ± Scm3 depleted and *cdc15as* pile-ups of contact maps of the 500 kb (top) and 200 kb (bottom) *cis* peri-centromeric regions (bin = 5 kb). (**d**) Log2 ratio between *cdc15as* Mcd1 ± Scm3 depleted and *cdc15as* pile-ups of contact maps of the 100 kb *trans* peri-centromeric regions (bin = 5 kb).

The online version of this article includes the following source data and figure supplement(s) for figure 6:

**Source data 1.** One blot shown in main *Figure 6a*.

**Figure supplement 1.** Cohesin organises centromere regions in telophase-arrested cells.

Based on evidence that suggested a role for cohesin during segregation, we sought to investigate whether a small pool of cohesin was retained on chromosomes in telophase. Interestingly, we detected binding to centromeres in this late mitotic stage (*Figure 5a and c*). Currently, we do not know whether the centromeric pool of cohesin bound in cdc15 cells is protected from separase cleavage, like centromeric cohesin during the first meiotic division, or loaded de novo during telophase. However, this pool of cohesin is actively organising, directly or indirectly, centromeric regions since depletion of cohesin in telophase revealed that bound cohesin during this late mitotic stage influences the structural organisation of pericentromeric regions, as well as promoting interactions between *CEN* regions of different chromosomes (*Figure 6b–d*). Further to this role at centromeric regions, we observed condensation defects at the ribosomal gene array on chromosome XII when we inactivated cohesin function in telophase-arrested cells (*Figure 7a*), demonstrating that the complex also plays a role in the organisation of this genomic site during late mitosis.

Interestingly, the loss of cohesin-dependent structures on chromosome arms, but not centromeric regions, on cdc15-arrested cells is reminiscent of the pattern reported for the inactivation of the cohesin regulator Pds5 (*Costantino et al., 2020*). Pds5 has been shown to restrict loop expansion (*Costantino et al., 2020*; *Dauban et al., 2020*) and therefore Pds5 absence in telophase could potentially explain our observations. In such scenario cohesin complexes loaded on centromeric regions might be able to extrude over long distances.

Currently, the functions of yeast cohesin during mitosis are thought to include: (i) the pairing of sister chromatids (*Guacci et al., 1997*; *Michaelis et al., 1997*), (ii) the bipolar organisation of chromatids as they attach to the mitotic spindles (*Tanaka et al., 2000*), and (iii) the organisation of metaphase chromosomes into looped domains (*Lazar-Stefanita et al., 2017*; *Schalbetter et al., 2017*). Recent work has shown that cohesin is necessary for the maintenance of the structure of the rDNA during metaphase and that this role is executed through the regulation of condensin localisation (*Lamothe et al., 2020*). Importantly, these cohesin functions occur in metaphase cells and separase cleavage of cohesin at the anaphase onset has generated the perception that cohesin's roles are finished at the beginning of anaphase. Collectively, our data demonstrates that yeast cohesin plays important roles during segregation, possibly after separase cleavage, maintaining the looped organisation of segregating chromatids and supporting centromere organisation. Therefore, these findings extend the repertoire of cohesin roles on chromosomes, demonstrating that this complex has important post-metaphase functions critical for ensuring faithful genome segregation.

## Methods

### Yeast strain and primers

Yeast strains used in this study are listed in *Table 1*. Epitope tagging of genes were carried out as described in *Janke et al., 2004*. Primers used in this study are described in *Table 2*.

### Media, culture conditions, and DNA constructs

To arrest the cells in G1, α-factor was added to exponentially growing MATa cultures (OD$_{600}$=0.5) to a final concentration of $3\times10^{-8}$ M for 3 hr at 25°C. To arrest cells in G2/M, nocodazole (1.5 mg/mL stock in DMSO 100%) was added to cultures with OD$_{600}$=0.5 to a final concentration of 0.015 mg/mL for 2.5 hr. To arrest cells in metaphase (Cdc20), cells carrying the *CDC20* gene under methionine repressible promoter MET3 (*MET3-CDC20*) were grown overnight in minimal media lacking methionine. The

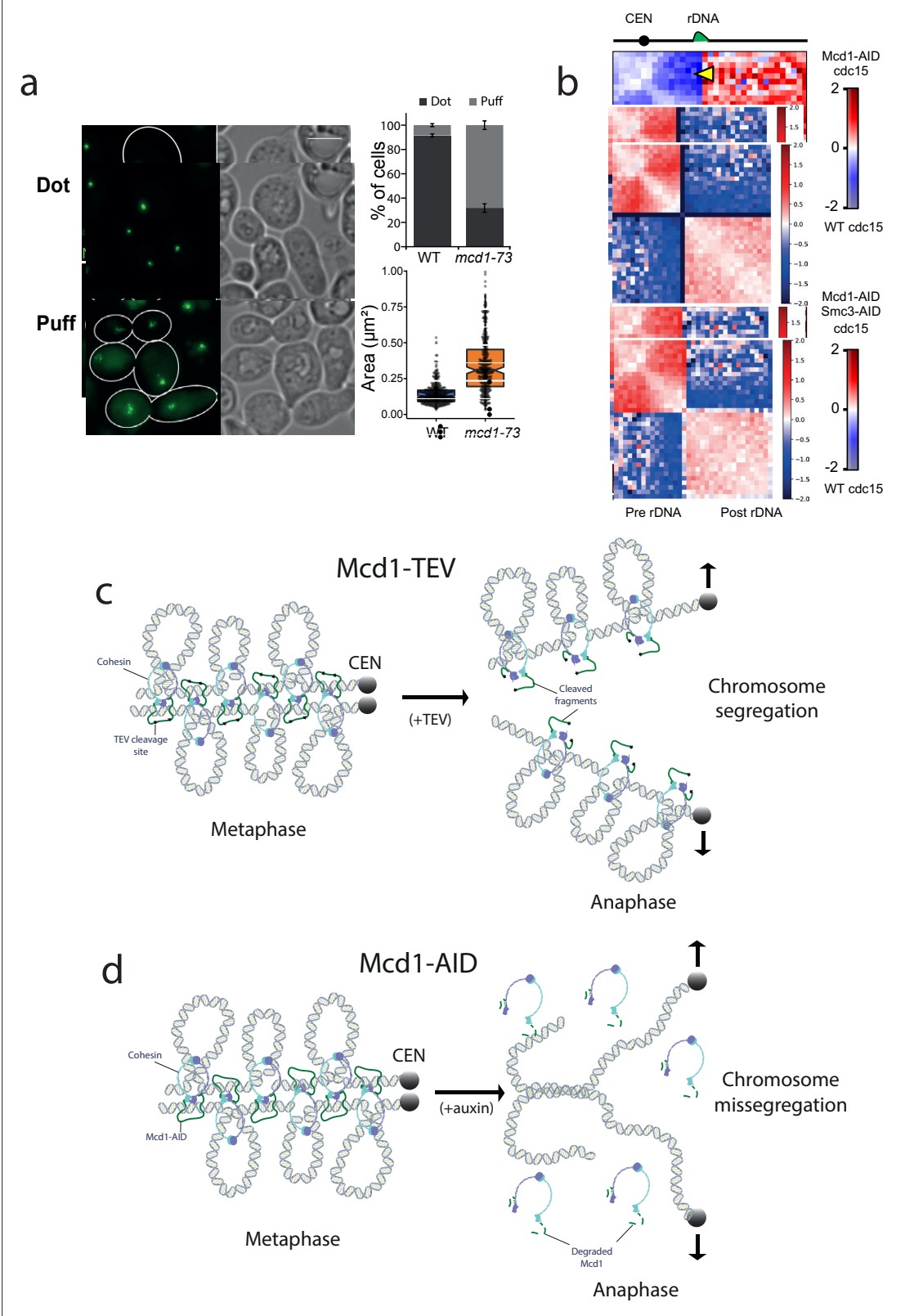

**Figure 7.** Cohesin has a structural role after metaphase. (**a**) Cells containing *cdc15-as* and *NET1*-yeGFP with either MCD1 wild-type (WT) or the temperature-sensitive allele *mcd1-73* were arrested in telophase (*cdc15-as*) at 25°C and then the temperature was shifted to 37°C for 30 min. Cells were then imaged under the microscope. Representative images of the experiment are shown. White scale bars represent 5 µm (left). Quantification of the rDNA morphology was scored (upper right). Net1-GFP area marking the rDNA was quantified (lower right). At least 100 cells of two biological replicas

*Figure 7 continued*

were quantified for each condition. (**b**) Log2 ratio of chromosome XII contact maps between Cdc15 and Cdc15 Mcd1-AID (top) or Cdc15 and Cdc15 Mcd1-AID Smc3-AID (bottom). The yellow arrow points at the contacts made between the *CEN12* and the rDNA. (**c**) Diagram showing a potential model explaining how TEV-cleaved cohesin could maintain the looped organisation of segregating chromatids. TEV cleavage of Mcd1 could be retained in one of the two segregating chromatids maintaining some of the structural functions. In this scenario sister chromatid cohesion would be lost but the loop organisation of individual chromatids would be partially maintained, thus facilitating segregation. (**d**) Diagram showing a potential model explaining how degradation of cohesin subunit Mcd1 could lead to catastrophic segregation. Mcd1 degradation leads to the destabilisation of cohesin on chromatin. In the absence of cohesin, though cohesion is dissolved (allowing genome separation), the loss of structure on separated chromatids would prevent their segregation and cells would exhibit the anaphase bridges phenotypes observed.

day after cells were arrested by washing the cells and resuspending them in rich media containing 5 mM methionine for 3 hr. To arrest cells in telophase, Cdc15 was tagged with an auxin degron (*CDC15-AID*), and IAA was added to the culture at a final concentration of 3 or 1 mM when growing in minimal media.

To arrest the cells in late telophase using the cdc15-as allele, cells released from an α-factor arrest were treated with 10 µM of the ATP analogue 1-NM-PP1 for 1 hr 45 min. Cultures were monitored by microscopy until ≥90% of cells were arrested. To release cells from G1, the culture was

**Table 1.** Yeast strains used in this study.

| | | |
|---|---|---|
| CCG14820 | CCG4000; *MET3-3HA-CDC20:TRP1; ADH1-OsTIR1::URA; MCD1:AID-9Myc:Hph; TetR-GFP::ADE2; TetO:469Kb ChrV:HIS3* | This study |
| CCG14881 | CCG4000; *MET3-3HA-CDC20:TRP1; ADH1-OsTIR1::URA; MCD1:AID-9Myc:Hph; TetR-GFP::ADE2; TetO:558Kb ChrV:HIS3* | This study |
| CCG14629 | CCG4000; *MET3-3HA-CDC20:TRP1; ADH1-OsTIR1::URA; MCD1:AID-9Myc:Hph; TetR-GFP::ADE2; TetO:448Kb ChrIV:HIS3* | This study |
| CCG14635 | CCG4000; *MET3-3HA-CDC20:TRP1; ADH1-OsTIR1::URA; MCD1:AID-9Myc:hph; TetR-GFP::ADE2; TetO:1513Kb ChrIV:HIS3* | This study |
| K9027 | *MATα; MCD1Δ:HIS3; MCD1TEV268::LEU2; GAL-NLS-mcy9-TEVprotease-NLS2::TRP1* (10-fold integrant by southern); *ura3::3xURA3 tetO112; his3::HIS3 tetR-GFP; MET3-HA3-CDC20::TRP1; ade2-1 can1-100 GAL psi+* | *Uhlmann et al., 2000* |
| CCG12955 | CCG12681; *SMC3-V5:HPH* | This study |
| CCG12356 | CCG4000; *MET3-3HA-CDC20:TRP1, FLAG-MCD1-268TEVF-6HA:LEU2, GAL- TEV-9Myc::URA3; tetR-GFP::ADE2* | This study |
| CCG12351 | CCG4000; *MET3-3HA-CDC20:TRP1, FLAG-MCD1-268TEVG-6HA:LEU2, GAL1-TEV-9Myc::URA3; tetR-GFP::ADE2* | This study |
| CCG14783 | CCG12351; *TetO:469Kb ChrV:HIS3* | This study |
| CCG14784 | CCG12356; *TetO:469Kb ChrV:HIS3* | This study |
| CCG14882 | CCG14635; *SMC3-V5:hph* | This study |
| CCG12925 | CCG12682; *SMC3-V5:HPH* | This study |
| CCG13574 | CCG4000, *MET3-3HA-CDC20:TRP1; pADH1-OsTIR1::URA; TetR-GFP::ADE2* | This study |
| CCG14635 | CCG4000; *MET3-3HA-CDC20:TRP1; ADH1-OsTIR1::URA; MCD1:AID-9Myc:hph; TetR-GFP::ADE2; TetO:1513Kb ChrIV:HIS3* | This study |
| CCG14880 | CCG4000; *ADH1-OsTIR1::LEU; CDC15:AID-9Myc:Hph* | This study |
| CCG14909 | CCG4000; *ADH1-OsTIR1::LEU; CDC15:AID-9Myc:Hph SMC3-yeGFP:Kan* | This study |
| CCG14910 | CCG4000; *ADH1-OsTIR1::LEU; CDC15:AID-9Myc:Kan; SMC3-yeGFP:Hph; SPC29-RedStar2:Nat* | This study |
| CCG1835 | *MATa bar1:hisG ura3-1 trp1-1 leu2-3,112 his3-11 ade2-1 can1-100 GAL+cdc15-2* | This study |
| CCG14731 | *CCG1835; MCD1-6HA:Hyg* | This study |
| CCG14758 | CCG4000; *SMC1-6HA:Nat; pADH1-OsTIR1::URA* | *Garcia-Luis et al., 2019.* |
| CCG14821 | CCG4000 *pADH1-OsTIR1::LEU; cdc15as1:KanMX* | This study |
| CCG14822 | CCG14821 *MCD1:AID-9Myc:Hph* | This study |
| CCG14823 | CCG14822 *SMC3:AID-9Myc:Hph* | This study |

**Table 2.** Primers used in this study.

| Primer number | Target | Primer |
|---|---|---|
| 4729 | Chr.V 235.1 (235000)_F | GCATGTGGATGTTTTTGGGGT |
| 4730 | Chr.V 235.1 (235125)_R | GAGACACTAGAGGGGCTATCCT |
| 4731 | Chr.V 443.1 (443017)_F | TCCACGTAATATTGCGGCCT |
| 4732 | Chr.V 443.1 (443151)_R | AATGCCAACTCAGCTTTGCG |
| 4733 | Chr.V 529.1 (529089)_F | ACTGAGAGGGAAGGACGACT |
| 4734 | Chr.V 529.1 (529198)_R | AATCGGCGTCAGACAAACCT |
| 4735 | Chr.V 548.3 (548276)_F | GGAAAATAGCCGCCCAAGGA |
| 4736 | Chr.V 548.3 (548398)_R | TGGCATAACAGACTACAGCAAA |
| 4737 | Chr.V 102.7(102733)_F | CGCATGCTTTTCTCAGACCTT |
| 4738 | Chr.V 102.7(102863)_R | TGCGGTACTGAGGGCCAAAT |
| 4739 | Chr.V 29.8(29844)_F | ATAGTTTGGGTGCTGCTGATT |
| 4740 | Chr.V 29.8(29970)_R | ACATTTTGCCGCCATACACA |
| 4557 | Ch.V_165.8_F | CGCGTTGGTCAAGCCTCATA |
| 4558 | Ch V_165.8_R | CACTACTCGGCTTCTTGCCA |
| 4563 | CEN5_152.2_F | CAAGCCACTGTTGGCGTTTC |
| 4564 | CEN5_152.2_R | TTATGTGCGGCTTTGTCAGC |

spun (4000 r.p.m., 1 min) and washed in YPD three times. The pellet was then resuspended in YPD containing 0.1 mg/mL pronase. To release cells from nocodazole, the culture was spun (4000 r.p.m., 1 min) and washed in YPD containing 1% DMSO five times. The pellet was then resuspended in YPD. To degrade proteins tagged with AID epitope, a stock of IAA of 0.6 M in ethanol 100% was used.

## Microscopy and statistics

To monitor cell cycle progression and chromosome segregation, an epifluorescence OLYMPUS IX70 microscope was used fitted with a Lumecor Spectra LED light source, a Hamamatsu Orca Flash 4.0 V2 camera and a ×100/1.35 lens. One mL of cell culture was taken from each timepoint and mixed with glycerol (20% final concentration) to preserve *TetO*/TetR signal after being frozen at –80°C. For visualisation, cells were centrifuged at 3000 r.p.m. for 2 min and 1 μL of the pellet was mixed with 1 μL of DAPI solution (DAPI 4 μg/mL Triton 1 %) on the microscope slide. For each field 20 Z-focal planes images were captured (0.3 μm depth between each consecutive image). Images were analysed with Fiji (*Schindelin et al., 2012*). To quantify the distance between the GFP dots, a Fiji macro was developed to automatically compute the weighted centroid of the dots and measure the three-dimensional distance between them. To visualise Smc3-yeGFP, cells were imaged fresh in a DELTAVISION Elite fluorescence microscope fitted with a Lumecor Spectra LED light source, a Photometrics Coolsnap HQ camera and a ×100/1.4 lens. At least 80 cells were quantified for each timepoint. Smc3-yeGFP signal intensity was calculated as the ratio of signal at the centromeres and the signal of the same area in the nucleus on Z-projection of images taken every 0.2 μm in 6 μm.

## Western blot

Protein extraction was done by lysing the cells in a FastPrep FP120 (BIO101) machine with 20% TCA and glass beads. Three repetitions of a 20 s cycle, power setting 5.5. Proteins were precipitated with TCA 7.5% and centrifuging at 15,000 r.p.m. for 10 min at 4°C. Then, the pellet was resuspended in Laemmli buffer ×1.5. Western blots were resolved in 7.5% SDS-PAGE gels. Proteins were transferred to polyvinylidene fluoride membranes using the TE70X semidry blotter system (Hoefer). The antibodies used were anti-HA (Roche, 3F10), anti-Myc (Roche, 9E10), anti-PGK1 (Thermo Fisher Scientific, 459250) anti-V5 (Abcam, ab9116), and anti-FLAG (Invitrogen MA1-142). Blots were incubated with the ECL Prime Western blotting detection reagent (GE Healthcare). Blots were developed by exposure

to high-performance chemiluminescence films (Amersham Hyperfilm ECL, GE Healthcare) or in an ImageQuant LAS 4000 mini machine (GE Healthcare).

## Chromatin immunoprecipitation

For ChIP analysis, cells were grown to $OD_{600} = 0.5$ and arrested at the required cell cycle stage. A total of 100 $OD_{600}$ units of *Saccharomyces cerevisiae* were collected. Cells were fixed for 15 min at 25°C and quenched with glycine (final concentration 125 mM) for 7 min before cells were harvested by centrifugation at 4000 r.p.m. for 1 min. The cell pellets were washed in PBS and transferred to a screw cap tube and frozen on dry ice. The pellets were stored at −80°C. Pellets were resuspended in 300 µL of IP buffer (150 mM NaCl, 50 mM Tris-HCl [pH 7.5], 5 mM EDTA, NP-40 [0.05% vol/vol], Triton X-100 [1% vol/vol]) containing PMSF (final concentration 1 mM) and complete protease inhibitor cocktail (without EDTA, from Roche). A 500 µL volume of glass beads was added to the tubes. Cells were broken in a FastPrep FP120 cell disruptor (BIO101) by three repetitions of a 20 s cycle at power setting 5.5. The cell lysate was transferred to a new tube and 100 µL volume of IP buffer containing PMSF and protease inhibitors was added. The cell lysate was spun down for 10 min at 15,000 r.p.m. at 4°C. This pellet was resuspended in 1 mL of IP buffer containing PMSF and protease inhibitors, and sonicated for 30 (30 s on, 30 s off) at high power at 4°C in a Diagenode Bioruptor pico. After sonication samples were spun down for 10 min at 15,000 r.p.m. the supernatant was taken. A 200 µL volume of the sonicated chromatin was taken as 'input' and 400 µL was incubated with 40 µg of anti-V5 antibody (anti-V5 Abcam, ab9116) in a sonicator at low power for 30 min (30 s on, 30 s off). The 'input' DNA was precipitated with 0.3 M sodium acetate and 2.5 volumes of cold ethanol and spun down at 15,000 r.p.m. for 30 min, then the supernatant was removed. The pellet was washed with 70% ethanol and air-dried. After antibody binding, the IP sample was spun down at 13,000 r.p.m. for 5 min and the supernatant was added to 60 µL of Dynabeads protein G (Invitrogen), previously equilibrated with IP buffer. The samples were then incubated for 2 hr at 4°C in a rotating wheel and washed five times with IP buffer using a magnetic separator rack. Finally, 'input' samples and IP samples were resuspended in de-crosslinking buffer (TE ×1, 1% SDS, 10 µg ml$^{-1}$ RNase A, 1 mg ml$^{-1}$ proteinase K) and incubated at 65°C overnight. Samples were purified using a ChIP DNA Clean & Concentrator kit (Zymoresearch) according to the manufacturer's instructions. Calibrated ChIP-seq were done as described in *Garcia-Luis et al., 2019*.

## Protein co-immunoprecipitation

One-hundred and twenty $OD_{600}$ of asynchronous cells were $OD_{600} = 1$ were collected and washed in cold water and resuspended in 200 µL of ice-cold buffer A (50 mM HEPES, 150 mM KCl, 1.5 mM MgCl$_2$, 0.5 mM DTT, and 0.5% Triton X-100 [pH 7.5] supplemented with complete protease inhibitor cocktail tablets, Roche). Five-hundred mL of glass beads (425–600 µm) were added and cells lysed in a FastPrep FP120 cell disruptor (BIO101) by three repetitions of a 20 s cycle at power setting 5.5. Extracts were maintained on ice for 2 min after each cycle. Cell extracts were centrifuged for 10 min at 12,000 r.p.m. at 4°C and the supernatant incubated with protein G Dynabeads (Invitrogen) bound to anti-Myc antibody (Roche, 9E10) for 2 hr at 4°C. Finally, beads were washed five times in washing buffer (10 mM Tris-Cl pH 7.5, 150 mM NaCl, 0.5% Triton) and unbound from the antibody by incubating at 37°C for 4 min in SR buffer (2% SDS, 0.125 M Tris-Cl, pH 6.8). Immunoprecipitated proteins were mixed with SS buffer (5% saccharose, 0.0125% bromophenol blue) and run in an SDS-PAGE gel.

## Hi-C libraries

Cells were fixed with 3% formaldehyde (F8775, Sigma-Aldrich) as detailed in *Dauban et al., 2020*. Formaldehyde was quenched with 300 mM of glycine at room temperature for 20 min. Hi-C experiments were performed with an Hi-C kit (Arima Genomics) involving a double DpnII+HinfI restriction digestion. Preparation of the samples for paired-end sequencing on an Illumina NextSeq500 (2×35 bp) was done with Collibri ES DNA Library Prep Kit for Illumina Systems (Thermo Fisher Scientific, A38605024).

## Generation and normalisation of contact maps

Alignment of the reads and processing of the contact data was done with Hicstuff using the S288C reference genome (*Matthey-Doret et al., 2020*). Hicstuff pipeline was launched with the following

parameters: aligning with bowtie2, filtering out spurious 3C events and PCR duplicates based on read positions. The 'view' mode of Hicstuff with the ICE function were used to generate normalised contact maps as described in *Imakaev et al., 2012*. Contact maps were binned at 50 kb for the whole genome or 1 kb for single chromosomes, and 30 kb for single chromosome ratio map.

### Contact probability as a function of the genomic distance p(s)

Genome-wide contact probability as a function of genomic distance pc(s) and its derivative were computed using the 'distance law' function of Hicstuff with the following parameters: removing 100 kb on both sides of the centromeres, averaging the contact data of each chromosome arms, removing the chromosome XII (*Matthey-Doret et al., 2020*).

### Aggregated of *cis* and *trans* centromere contacts

*Cis*-centromere pile-up contact maps are the result of averaged 205 or 505 kb windows centered on the 16 centromere positions (bin: 5 kb) generated with Chromosight in quantify mode (option: `--pattern border, --perc-zero=100`) (*Matthey-Doret et al., 2020*). These windows are taken from a balanced and detrended contact map. The detrending corrects for the distance-dependent contact decay due to polymeric behaviour (see *Matthey-Doret et al., 2020*, for more details). Because the contact map is symmetric and that the *cis*-centromere pile-up is centered on the diagonal, only half of the average pile-up is presented. *Trans*-centromere pile-up were similarly generated by Chomosight, but with the option `--inter`, and centered on the 120 centromere intersections.

## Acknowledgements

We thank members of our laboratories for discussion and critical reading of the manuscript. We thank Helle D Ulrich for kindly sharing the plasmids for auxin-inducible degron tagging. We also thank MRC-LMS microscopy facility for help with the microscope set-up and especially Chad Whilding for developing the Fiji macro for automated distance quantification of GFP dots. We thank Amaury Bignaud for his help with bioinformatics analysis.

The work in the Aragon laboratory was supported by the Medical Research Council (UKRI MC-A652-5PY00) and the Wellcome Trust (100955/Z/13/Z). This research was further supported by funding from The European Research Council (RK), Agence Nationale pour la Recherche (RK).

## Additional information

### Funding

| Funder | Grant reference number | Author |
| --- | --- | --- |
| Medical Research Council | UKRI MC-A652-5PY00 | Luis Aragon |
| Agence Nationale de la Recherche | | Romain Koszul |
| Wellcome Trust | 100955/Z/13/Z | Jonay Garcia-Luis |
| European Research Council | | Romain Koszul |

The funders had no role in study design, data collection and interpretation, or the decision to submit the work for publication. For the purpose of Open Access, the authors have applied a CC BY public copyright license to any Author Accepted Manuscript version arising from this submission.

### Author contributions

Jonay Garcia-Luis, Data curation, Formal analysis, Investigation, Writing - review and editing; Hélène Bordelet, Formal analysis, Investigation, Writing - review and editing; Agnès Thierry, Investigation; Romain Koszul, Resources, Formal analysis, Supervision, Funding acquisition, Writing - review and editing; Luis Aragon, Conceptualization, Supervision, Funding acquisition, Writing - original draft, Project administration, Writing - review and editing

## Author ORCIDs

Hélène Bordelet (iD) http://orcid.org/0000-0002-8190-2326
Romain Koszul (iD) http://orcid.org/0000-0002-3086-1173
Luis Aragon (iD) http://orcid.org/0000-0003-0634-6742

## Decision letter and Author response

Decision letter https://doi.org/10.7554/eLife.80147.sa1
Author response https://doi.org/10.7554/eLife.80147.sa2

---

# Additional files

## Supplementary files

• MDAR checklist

## Data availability

Sequencing data have been deposited in GEO under accession code GSE183481.

The following dataset was generated:

| Author(s) | Year | Dataset title | Dataset URL | Database and Identifier |
|---|---|---|---|---|
| Garcia-Luis J, Bordelet H, Thierry A, Koszul R, Aragon L | 2021 | Cohesin contribution to chromatid organisation is critical during chromosome segregation | https://www.ncbi.nlm.nih.gov/geo/query/acc.cgi?acc=GSE183481 | NCBI Gene Expression Omnibus, GSE183481 |

The following previously published dataset was used:

| Author(s) | Year | Dataset title | Dataset URL | Database and Identifier |
|---|---|---|---|---|
| Garcia-Luis J, Lazar-Stefanita L, Gutierrez-Escribano P, Thierry A, Cournac A, García A, González S, Sánchez M, Jarmuz A, Montoya A, Dore M, Kramer H, Karimi MM, Antequera F, Koszul R, Aragon L | 2018 | FACT mediates cohesin function on chromatin | https://www.ncbi.nlm.nih.gov/geo/query/acc.cgi?acc=GSE118534 | NCBI Gene Expression Omnibus, GSE118534 |

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
