## [Editor Report]

Previous studies suggest that multiple activities of cohesin, essential in mitotic chromosome structure and segregation, are required only prior to the onset of chromosome segregation, but this study in *Saccharomyces cerevisiae*, using different alleles of the Mcd1 subunit, shows that cohesin plays also a role in anaphase organizing the centromeric regions, providing new evidence that cohesin function is critical during and after the onset of chromosome segregation. The work is of relevance to understanding chromosome biology and cell division.

---

## [Decision Letter]

**Decision letter after peer review:**

Thank you for submitting your article "Cohesin contribution to chromatid organisation is critical during chromosome segregation" for consideration by *eLife*. Your article has been reviewed by 3 peer reviewers, one of whom is a member of our Board of Reviewing Editors, and the evaluation has been overseen by Jessica Tyler as the Senior Editor. The reviewers have opted to remain anonymous.

Essential revisions:

1) Figure 6b and c. Why do the authors change tags when measuring chromatin-bound cohesin in metaphase vs telophase? Changing tags would appear to compromise a comparison between metaphase and telophase arrested cells.

2) Figure 7B left. Please explain why you present a log ratio with Cdc15 in the denominator when in all the other panels they present a log ratio of Cdc15 in the numerator the more intuitive ratio (condensed- more contacts /uncondensed – fewer contacts)? The authors do not mention the large blue squares or flares. More interpretation of the findings of this figure would be helpful.

3) Figure 7B right. This reader's interpretation is that the white squares associated with each chromosome mean that the chromosome-wide contacts in cdc15 arrested cells (compared to G1) are unchanged by inactivation by Mcd1. The flairs in cdc15 arrested cells (compared to G1) seem enhanced. Thus, it seems from their data that extended interactions from the centromere exist in cdc15 arrested cells that are further enhanced by removing most of the cohesin. Please clarify.

4) 7C and 7D. It is not clear what the first three panels of these panels show.

5) Figure 8. The authors should guide the reader through this Hi C in this figure.

6) Figure 8a. Why do the authors switch to using the mcd1-73 in this experiment instead of the MCD1-AID auxin depletion that they used in all the other experiments?

The reasons for the change in reagents should be described new reagents should be characterized.

7) The first page and half reads like a review and should be deleted.

8) The authors claim that figure 1 shows the degradation of cohesin in anaphase leads to catastrophic segregation. This overstates the experimental results. What they show is the inactivation of cohesin in a metaphase arrested cell causes false anaphase with catastrophic segregation. While consistent with a function during anaphase, they cannot rule out from this experiment that cohesin function in chromosome structure is only required metaphase (where the cells are arrested) and it is revealed by the false anaphase. Thus, the function for cohesin after metaphase is critically dependent upon the Hi-C analysis of cdc15 arrested cells.

9) Mounting evidence from in vivo experiments suggests that loop extrusion may not involve topological entrapment (see Dekker and Michael Peters). The authors provide evidence that this might also be the case in vivo based on the presence of loops in the TEV cleavage experiment. However, this conclusion must be softened to include the following caveat. The pattern of expanded loops observed in cdc15 arrested cells is very reminiscent of Pds5 depleted cells. That pattern appears to reflect a very small amount of chromosome-bound cohesin that persists in the absence of Pds5. Thus, the authors cannot exclude that the loops they observe in cdc15 arrested cells or TEV induced cells are not due to a small amount of cohesin with uncleaved Mcd1. To make this conclusion the authors would have to show that all Mcd1p is cleaved by TEV.

10) The method used to invoke a cdc20 shutoff to arrest cells in metaphase should be explained in Materials and methods.

11) The assays involving various GFP-marked chromosomes should be explained better. When the GFP-marked locus is very close to the centromere, missegregation only happens when the sister chromatids have attached to microtubules of the same spindle poles. In other words, a defect in kinetochore function. When the GFP marked locus is at a CEN distal locus, a missegregation is a combination of kinetochore defects and a failure to condense the chromosome enough to ensure the distal ends can clear the mid-zone of the cells. The missegregation due to condensation defects can be calculated by subtracting the missegregation from CEN proximal marked loci.

12) The assertion that cohesin is generated loops in anaphase to support compaction is a weakly supported aspect of the paper. Figure 5B doesn't show any obvious loop structures in either TEV-induced or auxin-induced anaphase. This could be because loop boundary factors are no longer functional in anaphase, obscuring any looping function. Evidence for residual looping along chromosome arms rests on the P(s) analysis in Figure 5C. This analysis shows increased contact frequency in the Mcd1-TEV relative to Mcd1-AID at lengths from 20kb to 200+kb. This is an analysis of all contacts, including those involving centromeres. Figure 7 shows that cohesin is required for centromere contact with regions all across the chromosome arm with little evidence of cohesin-dependent contacts along chromosome arms. The extensive cohesin-dependent contacts from the centromere could be biasing the analysis of Figure 5C. The P(s) analysis in figure 5C should be re-analyzed excluding contacts containing sequences in the centromeric and pericentromeric regions. This will give a more accurate assessment of potential looping along chromosome arms.

13) In figure 2C a mis-segregation analysis of a CEN5 tetO tag is performed. This tag is not examined in figure 1 so it does not give any relevant information for comparing mis-segregation in depleted Scc1 cells compared to artificially cleaved Scc1 cells. If the authors want to use this data, they should also show the analysis of the CEN5 tetO tag for depleted Scc1 cells.

14) The relevant comparison for comparing Mcd1-TEV relative to Mcd1-AID shown is the relative mis-segregation of the 469tetO tags, shown in figure 1C and figure 4C which have relative mis-segregation rates of <5% mis-segregation and 20% mis-segregation at the end of the experiment. But in figure 4C mis-segregation increases to >10% at 60 minutes before reducing to <5%. How can an apparent reversal of mis-segregation occur in this context?

15) Figure 1c,d, and 2b,c should show the SEM values for the points taken.

16) The authors start with the premise that all arm cohesin is destroyed in animals and plant cells by the prophase removal pathway. This is incorrect. In a normal unperturbed cell cycle, most of the cohesin is removed from chromosome arms but arms remained paired until the onset of anaphase implying that a small fraction of cohesin along the arms persists. Furthermore, substantial amounts of cohesin are reloaded on chromosomes by telophase. The purpose of this cohesin is unknown. The authors' results suggest that the presence of chromosome-bound cohesin post metaphase is conserved in yeast, and therefore is likely a general feature of all mitoses. The function of this chromosome-bound cohesin in telophase in other eukaryotes was unknown. The results from this manuscript provide a potentially important clue.

17) The failure to constrain centromere anchored loops upon inactivation of most cohesin is reminiscent of loops seen in pds5 mutants. The authors should cite this result and incorporate it into a better discussion of condensation in anaphase.

18) The results may also connect with meiosis where a subset of cohesin is protected from separase during meiosis. Authors may think of additional examples and bring it to Discussion.

19) The log ratio of G1 or Cdc15 contacts shows large chromosomal domains (blue squares) and flares (presumably condensation emanating from the centromeres). These types of domains have not been reported in previous analyses of nocodazole cells. They suggest a more and or distinct compacted state in Cdc15 cells. Interestingly this conclusion fits with the FISH of rDNA in cdc15 arrested cells which looks much more like that seen in most other eukaryotes. Authors should extend this in the Discussion.

---

## [Author Response]

Essential revisions:1) Figure 6b and c. Why do the authors change tags when measuring chromatin-bound cohesin in metaphase vs telophase? Changing tags would appear to compromise a comparison between metaphase and telophase arrested cells.

We thank the reviewer for raising this point. From the comment it is clear that we failed to explain the experiment correctly in our manuscript. In this experiment, we performed ChiP analysis of Mcd1 (Mcd1-6HA) in telophase arrested cells only (using the *cdc15-2* allele). We had two conditions, one with cells carrying a tagged Mcd1 (Mcd1-6HA) allele and another where Mcd1 was untagged. We used the untagged signals to subtract the background signal, thus reveal the binding due to Mcd1-6HA in telophase arrested cells.

We used the localisation of SMC1-6HA in metaphase arrested cells only to illustrate the normal position of cohesin complexes in metaphase cells (a time when they are most abundant on chromosomes), as a reference to be compared to our analysis of Mcd1 binding at telophase. The localisation of SMC1-6HA in metaphase was not meant to be qualitatively compared to Mcd1-6HA ChIP peaks in our telophase arrests. For SMC1-6HA we used a dataset already created and published previously in the lab (GSE118534). We have described the experiment in more detail in the figure legend to ensure that there is no misunderstanding.

2) Figure 7B left. Please explain why you present a log ratio with Cdc15 in the denominator when in all the other panels they present a log ratio of Cdc15 in the numerator the more intuitive ratio (condensed- more contacts /uncondensed – fewer contacts)? The authors do not mention the large blue squares or flares. More interpretation of the findings of this figure would be helpful.

The reviewer is correct to point out this discrepancy. To avoid any confusion and facilitate comparisons we now show all ratio maps with cdc15 WT always presented as the denominator. Regarding the “large blue squares” which appeared on the cdc15 vs. G1 map, they reflect compaction of chromosomes during late anaphase with respect to G1. However, we have now removed the left panel 7b comparing the contact maps of G1 vs cdc15 arrested cells. Indeed, we believe this is not an observation directly relevant to the present work, which aims at assessing the role of cohesin after metaphase. Also, cdc15 HiC maps were reported before and their compaction shown to mostly dependent on condensins (Lazar-Stefanita et al., 2017).

3) Figure 7B right. This reader's interpretation is that the white squares associated with each chromosome mean that the chromosome-wide contacts in cdc15 arrested cells (compared to G1) are unchanged by inactivation by Mcd1. The flairs in cdc15 arrested cells (compared to G1) seem enhanced. Thus, it seems from their data that extended interactions from the centromere exist in cdc15 arrested cells that are further enhanced by removing most of the cohesin. Please clarify.

We removed the comparison with G1 (above) but overall, the reader’s interpretation is indeed correct to state that cohesion depletion has little effect on most cdc15 chromosomes but chr. XII.

We now discuss the two main points below:

First, the immediate depletion of cohesin subunits in cdc15 arrested cells has the major effect of altering the organisation of chromosome XII (discussed in the figure 8), as well as the contacts made by centromeric regions.

On a large scale (above 200 kb), cohesins favour cis contacts between centromeres and their chromosome arms. However, this effect is reversed at short distances (<100 kb). Indeed, pannels 7c show that cohesin impedes contacts between centromeres and their ~100 kb flanking regions. To illustrate this scaling effect, we now added a pileup contact maps of 500 kb windows centered on centromeres that shows both effects. We have also highlighted this description in the text.

4) 7C and 7D. It is not clear what the first three panels of these panels show.

Sorry for the lack of clarity. These panels are actually not that informative and have been moved to the supplementary data while we kept in the main figure more detailed representations of the ratio between the plots. Briefly, the first three panels of Figure 7c represented the average of 16 x 505 kb windows centered on the 16 centromeres on the diagonal of normalized and detrended contact maps. The normalization of the contact map attenuates the experimental biases, and the detrending corrects for the distance-dependent contact decay due to polymeric behavior (note that because of the symmetry of the contact map, only half of the average cis-centromere pileup was shown). For Figure 7d, these panels corresponded to trans contacts pile-ups.

We modified and detailed the methods to bring clarifications:

“Cis-centromere pile-up contact maps are the result of averaged 205 kb or 505 kb windows centered on the 16 centromere positions (bin: 5 kb) generated with Chromosight in quantify mode (option: --pattern border, --perc-zero=100) (Matthey-Doret et al., 2020). These windows are taken from a balanced and detrended contact map. The detrending corrects for the distance-dependent contact decay due to polymeric behaviour (see Matthey-Doret et al., 2020 for more details). Because the contact map is symmetric and that the cis-centromere pileup is centered on the diagonal, only half of the average pileup is presented. Trans-centromere pileup were similarly generated by Chomosight, but with the option --inter, and centered on the 120 centromere intersections. “

5) Figure 8. The authors should guide the reader through this Hi C in this figure.

To facilitate the description and interpretation of Figure 8 Hi-C data, we have added didactic elements to panel b.

First, we detail the figure 8b with more details in the main text to better guide the reader:

"In Cdc15-arrested cells, condensin-dependent loop bridges the CEN12 and rDNA (LazarStefanita et al., 2017). In cohesin-depleted cdc15-arrested cells, these contacts are also strongly decreased (Figure 8b, yellow arrow), suggesting that cohesin plays a role in the formation of this structure. Furthermore, contacts between pre- and post-rDNA regions are increased when the cohesin complex is disrupted, suggesting that it promotes the isolation of these regions in cdc15-arrested cells (Figure 8b). In conclusion, these results further support an active role for cohesin in the organisation of specific regions of the genome in telophase-arrested cells. "

6) Figure 8a. Why do the authors switch to using the mcd1-73 in this experiment instead of the MCD1-AID auxin depletion that they used in all the other experiments?The reasons for the change in reagents should be described new reagents should be characterized.

We thank the reviewers for raising this point. The reason behind this change is purely technical. We have consistently observed that degradation of AID tagged proteins is not efficient when cells are at 37°C. Therefore, we felt that using the genotype *cdc15-2 MCD1AID* where we would need to raise the temperature to 37°C initially before inducing AIDmediated degradation of Mcd1 might generate misleading results due to the inefficiency of AID degradation at this temperature.

In order to stably arrest cells in telophase and then compromise Mcd1 function we thought we could switch to Cdc15-AID mcd1-73, where we could achieve efficient AID mediated degradation (of Cdc15) initially at 25°C, and once this degradation had been achieved and cells were stably arrested, we could then raise the temperature to inactivate Mcd1 (using the mcd1-73 allele).

7) The first page and half reads like a review and should be deleted.

The first page and half is the introduction of the manuscript. Completely removing the whole introduction would make it difficult for readers. Therefore, although we have not completely removed this section, we have changed the style to ensure that this does not read like a review.

8) The authors claim that figure 1 shows the degradation of cohesin in anaphase leads to catastrophic segregation. This overstates the experimental results. What they show is the inactivation of cohesin in a metaphase arrested cell causes false anaphase with catastrophic segregation. While consistent with a function during anaphase, they cannot rule out from this experiment that cohesin function in chromosome structure is only required metaphase (where the cells are arrested) and it is revealed by the false anaphase. Thus, the function for cohesin after metaphase is critically dependent upon the Hi-C analysis of cdc15 arrested cells.

We agree with this reviewer. It has been described in the literature that cleavage of Mcd1 using TEV leads to TEV-induced anaphases, this has been used in a number of studies to investigate processes that occur in anaphase. However, we agree that in these induced anaphases cells are technically arrested in metaphase. Our degradation of Mcd1 in Figure 1 would be equivalent to TEV-induced anaphases and therefore carries the caveat highlighted by this reviewer. We agree that the strongest argument for the role of cohesion in anaphase is dependent on our HiC data. In order to ensure that we do not overstate the results we now state that auxin-mediated degradation of Mcd1 leads to “impaired separation of chromatids”.

9) Mounting evidence from in vivo experiments suggests that loop extrusion may not involve topological entrapment (see Dekker and Michael Peters). The authors provide evidence that this might also be the case in vivo based on the presence of loops in the TEV cleavage experiment. However, this conclusion must be softened to include the following caveat. The pattern of expanded loops observed in cdc15 arrested cells is very reminiscent of Pds5 depleted cells. That pattern appears to reflect a very small amount of chromosome-bound cohesin that persists in the absence of Pds5. Thus, the authors cannot exclude that the loops they observe in cdc15 arrested cells or TEV induced cells are not due to a small amount of cohesin with uncleaved Mcd1. To make this conclusion the authors would have to show that all Mcd1p is cleaved by TEV.

We thank this reviewer for raising this point. We ensured that full length Mcd1 was not detectable in the TEV-anaphase samples taken and used for HiC analysis (as shown in Suppl. Figure 4). However, it is possible that a very small fraction (not detectable by western) of Mcd1 escaped TEV cleavage. We will point out in the manuscript that although. Therefore, this should soften this argument and will include the possibility raised by the reviewer.

10) The method used to invoke a cdc20 shutoff to arrest cells in metaphase should be explained in Materials and methods.

We acknowledge that it would be important to include a detailed description of the arrest. We have now described the MET3-Cdc20 in more detail in the methods section.

11) The assays involving various GFP-marked chromosomes should be explained better. When the GFP-marked locus is very close to the centromere, missegregation only happens when the sister chromatids have attached to microtubules of the same spindle poles. In other words, a defect in kinetochore function. When the GFP marked locus is at a CEN distal locus, a missegregation is a combination of kinetochore defects and a failure to condense the chromosome enough to ensure the distal ends can clear the mid-zone of the cells. The missegregation due to condensation defects can be calculated by subtracting the missegregation from CEN proximal marked loci.

We thank this reviewer for the suggestion. We agree that a better explanation for the causes leading to missegregation of different tags would be helpful to the reader to better understand the defects. We have done the calculation suggested using centromere proximal and distal tags on chromosome IV, the longest chromosome not bearing the rDNA. The new data has been included in the supplementary materials.

12) The assertion that cohesin is generated loops in anaphase to support compaction is a weakly supported aspect of the paper. Figure 5B doesn't show any obvious loop structures in either TEV-induced or auxin-induced anaphase. This could be because loop boundary factors are no longer functional in anaphase, obscuring any looping function. Evidence for residual looping along chromosome arms rests on the P(s) analysis in Figure 5C. This analysis shows increased contact frequency in the Mcd1-TEV relative to Mcd1-AID at lengths from 20kb to 200+kb. This is an analysis of all contacts, including those involving centromeres. Figure 7 shows that cohesin is required for centromere contact with regions all across the chromosome arm with little evidence of cohesin-dependent contacts along chromosome arms. The extensive cohesin-dependent contacts from the centromere could be biasing the analysis of Figure 5C. The P(s) analysis in figure 5C should be re-analyzed excluding contacts containing sequences in the centromeric and pericentromeric regions. This will give a more accurate assessment of potential looping along chromosome arms.

Absolutely right. To support the point that cohesin participates directly or indirectly in the structuring of non-centromeric regions we now present the P(s) where 100 kb of regions on both sides of the centromeres are removed, each chromosome arm is considered separately, and chromosome XII is removed (three steps we also include to avoid polymer-brush like potential influence on contacts). The Materials and methods section for the P(s) calculation was modified accordingly.

Despite the exclusion of the pericentromeric regions, contacts between 20 and 200 kb remain more enriched in the Mcd1-TEV condition than in the Mcd1-AID condition suggesting that the cleaved cohesin complex still contribute, directly or indirectly, to the structuration of chromosome arms after metaphase.

13) In figure 2C a mis-segregation analysis of a CEN5 tetO tag is performed. This tag is not examined in figure 1 so it does not give any relevant information for comparing mis-segregation in depleted Scc1 cells compared to artificially cleaved Scc1 cells. If the authors want to use this data, they should also show the analysis of the CEN5 tetO tag for depleted Scc1 cells.

We thank the reviewer for this raising this point. In Figure 2c used the original strain used by the Nasmyth lab (in Uhlmann et al., 2000) to confirm previous results published. However, we understand the logic of the criticism and thus we have moved this figure to Supplementary Figure 2.

14) The relevant comparison for comparing Mcd1-TEV relative to Mcd1-AID shown is the relative mis-segregation of the 469tetO tags, shown in figure 1C and figure 4C which have relative mis-segregation rates of <5% mis-segregation and 20% mis-segregation at the end of the experiment. But in figure 4C mis-segregation increases to >10% at 60 minutes before reducing to <5%. How can an apparent reversal of mis-segregation occur in this context?

We understand the concern expressed by this reviewer regarding the kinetics of chromosome missegregation in Figure 4C. In TEV induced anaphases (Figure 4C), by 60 minutes of TEV protease induction, part of the population has entered DNA bulk segregation as detected by DAPI (as can be observed in Figure 4b-anaphase category). At this timepoint chromosome tags, specially those located nearer to the telomeres have not segregated yet and consequently are scored as missegregated. At later timepoints some of these cells seem to manage to resolve and/or segregate the chromosome tags correctly and therefore the level of missegregation is reduced at later timepoints. We hope this explains the apparent reverasal of missegregation. We note that his is not observed in Mcd1-AID (Figure 1).

15) Figure 1c,d, and 2b,c should show the SEM values for the points taken.

We understand the concern expressed by this reviewer. We have now included the SEM values in these figures.

16) The authors start with the premise that all arm cohesin is destroyed in animals and plant cells by the prophase removal pathway. This is incorrect. In a normal unperturbed cell cycle, most of the cohesin is removed from chromosome arms but arms remained paired until the onset of anaphase implying that a small fraction of cohesin along the arms persists. Furthermore, substantial amounts of cohesin are reloaded on chromosomes by telophase. The purpose of this cohesin is unknown. The authors' results suggest that the presence of chromosome-bound cohesin post metaphase is conserved in yeast, and therefore is likely a general feature of all mitoses. The function of this chromosome-bound cohesin in telophase in other eukaryotes was unknown. The results from this manuscript provide a potentially important clue.

We thank this reviewer for this comment. We have included these points in the introduction of our revised manuscript.

17) The failure to constrain centromere anchored loops upon inactivation of most cohesin is reminiscent of loops seen in pds5 mutants. The authors should cite this result and incorporate it into a better discussion of condensation in anaphase.

We agree with this reviewer and have cited this work and included the points on the Discussion section.

18) The results may also connect with meiosis where a subset of cohesin is protected from separase during meiosis. Authors may think of additional examples and bring it to Discussion.

We have mentioned the protection of cohesion from separase cleavage in meiosis in the discussion.

19) The log ratio of G1 or Cdc15 contacts shows large chromosomal domains (blue squares) and flares (presumably condensation emanating from the centromeres). These types of domains have not been reported in previous analyses of nocodazole cells. They suggest a more and or distinct compacted state in Cdc15 cells. Interestingly this conclusion fits with the FISH of rDNA in cdc15 arrested cells which looks much more like that seen in most other eukaryotes. Authors should extend this in the Discussion.

We have removed this ratio map to simplify our message.

However, this is how we interpret these results: in Cdc15, chromosomes are more compact than in G1 (also shown in Lazar-Stefanita 2017), hence the blue squares when the ratio map is plotted. Also, centromeres will tend to be brought in contact with the chromosome arms in cis because of condensin mediated looping taking place (resulting in the “flares”): loops would expand DNA until encountering the centromere which would act as a roadblock.